# Microtubule networks in zebrafish hair cells facilitate presynapse transport and fusion during development

**Saman Hussain[1], Katherine Pinter[1], Mara Uhl[2,3], Hiu-Tung Wong[1], Katie S Kindt[1]***

[1]Section on Sensory Cell Development and Function, National Institute on Deafness and other Communication Disorders, Bethesda, United States; [2]Presynaptogenesis and Intracellular Transport in Hair Cells Junior Research Group, Institute for Auditory Neuroscience and InnerEarLab, University Medical Center Goettingen, Goettingen, Germany; [3]Collaborative Research Center 889 'Cellular Mechanisms of Sensory Processing', Goettingen, Germany

## eLife Assessment

This **fundamental** study provides new insights into the maturation of ribbon synapses in zebrafish neuromast hair cells. Live-cell imaging and pharmacological and genetic manipulations together provide **compelling** evidence that the formation of this synaptic organelle is a dynamic process involving the fusion of presynaptic elements and microtubule transport, though the evidence that ribbon precursors move in a directed motion toward the active zone is less persuasive. These findings will be of interest to neuroscientists studying synapse formation and function and should inspire further research into the molecular basis for synaptic ribbon maturation.

***For correspondence:**
katie.kindt@nih.gov

**Competing interest:** The authors declare that no competing interests exist.

**Abstract** Sensory cells in the retina and inner ear rely on specialized ribbon synapses for neurotransmission. Disruption of these synapses is linked to visual and auditory dysfunction, but it is unclear how these unique synapses form. Ribbon synapses are defined by a presynaptic density called a ribbon. Using live imaging in zebrafish hair cells, we find that numerous small ribbon precursors are present throughout the cell early in development. As development progresses, fewer large ribbons remain, and localize at the presynaptic active zone (AZ). Using tracking analyses, we show that ribbon precursors exhibit directed motion along an organized microtubule network to reach the presynaptic AZ. In addition, we show that ribbon precursors can fuse together on microtubules. Using pharmacology, we find that microtubule disruption interferes with ribbon motion, fusion, and normal synapse formation. Overall, this work demonstrates a dynamic series of events that underlies the formation of a critical synapse required for sensory function.

## Introduction

The inner ear and retina contain sensory cells with specialized ribbon synapses that faithfully transmit the timing, duration, and intensity of sensory stimuli to the brain. These synapses are critical for hearing, balance, and vision, and their disruption is linked to auditory, vestibular, and visual disorders (*Frederick and Zenisek, 2023*; *Kujawa and Liberman, 2015*; *Wan et al., 2019*). The hallmark feature of these synapses is the presynaptic ribbon, a dense body made up primarily of the protein Ribeye (*Schmitz et al., 2000*). Ribbons are found at the presynaptic AZ and act as scaffolds to ready synaptic vesicles for release (*Schmitz, 2009*). Work in fixed tissues has led to the hypothesis that small ribbon

precursors migrate to the AZ and fuse to form larger, mature ribbons. However, direct evidence for these dynamic processes during ribbon formation has not yet been demonstrated.

Neurotransmission at mature ribbon synapses is triggered in response to graded membrane depolarizations dictated by the duration and intensity of sensory stimuli. Membrane depolarization opens voltage-gated calcium channels (Ca$_V$1) beneath ribbons (*Brandt et al., 2003*; *Chang et al., 2006*) (see schematic in *Figure 1C*). Calcium influx triggers synaptic vesicle fusion and the release of glutamate onto postsynaptic receptors (*Obholzer et al., 2008*; *Ruel et al., 2008*). Studies have shown Ribeye, the core component of the ribbon, is essential for the formation and function of ribbon synapses in both mouse and zebrafish (*Becker et al., 2018*; *Jean et al., 2018*; *Lv et al., 2016*; *Maxeiner et al., 2016*). Other key components include the classic neuronal scaffolding proteins Bassoon and a novel variant of Piccolo, Piccolino (*Gundelfinger et al., 2015*). In mice, the loss of Bassoon disrupts ribbon anchoring and synapse function (*Dick et al., 2003*; *Frank et al., 2010*; *Khimich et al., 2005*). In rats, the loss of Piccolino impacts ribbon morphology (*Michanski et al., 2023*; *Regus-Leidig et al., 2014*). Currently, how these key molecular players fit within the dynamics underlying ribbon formation is unclear.

In neurons, precursor vesicles containing Piccolo and Bassoon are transported along microtubules to the developing presynaptic AZ. (*Ahmari et al., 2000*; *Gundelfinger et al., 2015*; *Maas et al., 2012*; *Shapira et al., 2003*). This transport requires molecular motors, along with adaptor proteins (*Bury and Sabo, 2011*; *Fejtova et al., 2009*; *Maas et al., 2012*). In neurons, kinesins are the molecular motors that transport cargo in the anterograde direction–towards the presynaptic AZ—while cytoplasmic dynein mediates retrograde transport, back to the cell soma (*Sweeney and Holzbaur, 2018*). Although the kinesin responsible for Piccolo-Bassoon vesicle transport is not known, recent work in *Drosophila* and *C. elegans* suggests that the kinesin KIF1A may transport AZ components (*Oliver et al., 2022*; *Pack-Chung et al., 2007*). Work in developing photoreceptors and mouse auditory inner hair cells (IHCs), has shown that ribbon precursors contain not only Ribeye, but also Bassoon and Piccolino (*Regus-Leidig et al., 2009*; *Michanski et al., 2019*). Whether ribbon precursors are actively transported along microtubules during synapse formation is not known.

The formation of ribbon synapses has been extensively studied using light and electron microscopy in fixed tissues (*Michanski et al., 2019*; *Regus-Leidig et al., 2009*; *Schmitz, 2009*; *Sheets et al., 2011*; *Sobkowicz et al., 1986*; *Sobkowicz et al., 1982*). In mouse auditory IHCs, this process occurs over an extended time period (E18-P14) (*Michanski et al., 2019*; *Sobkowicz et al., 1986*; *Sobkowicz et al., 1982*), while in zebrafish hair cells, ribbon synapses mature in just 12–18 hr (*Dow et al., 2015*; *Sheets et al., 2011*). Early development in both mouse IHCs and zebrafish hair cells features many small ribbon precursors throughout the cell, likely formed via Ribeye self-aggregation in the cytosol (*Magupalli et al., 2008*; *Schmitz et al., 2000*). As development progresses, ribbons enlarge, localize to the presynaptic AZ, and associate with the innervating afferent terminals. Finally, the number of ribbons associated with postsynaptic machinery is refined to obtain the proper number of complete synapses. Recent work in mice has shown that ribbon precursors associate with microtubules (*Michanski et al., 2019*), and it has been proposed that ribbon precursors may migrate along microtubules to reach the presynaptic AZ, although the in vivo dynamics of this process remain unclear.

To study ribbon formation, we examined hair cells and developing ribbons in the zebrafish lateral line (*Figure 1A*), a sensory system that allows aquatic vertebrates to sense local water movements (*Freeman, 1928*; *Suli et al., 2012*). The lateral line consists of clusters of hair cells called neuromasts that are arranged in lines along the surface of the fish. In the posterior lateral line (pLL), which forms an array of neuromasts along the zebrafish trunk, hair cells emerge at 2 days post fertilization (dpf) (see *Figure 1A–B and D*), and the lateral-line system reaches functional maturity by 5 dpf (*Suli et al., 2012*). At these larval ages, zebrafish are transparent, which enables in vivo imaging of ribbon formation over extended periods (*Dow et al., 2015*). Further, transgenic lines expressing fluorescently tagged proteins allow high-resolution visualization of subcellular dynamics, including ribbon formation.

During our study, we worked in close collaboration with another group investigating the late stages of ribbon formation in mouse auditory IHCs (postnatally). This work is published in a companion paper (*Voorn et al., 2024*). Together, our studies demonstrate that ribbon transport along microtubule networks is essential for proper ribbon formation in mice and zebrafish. Our zebrafish work leverages transgenic lines that label: developing ribbons, microtubule networks, and the growing plus ends

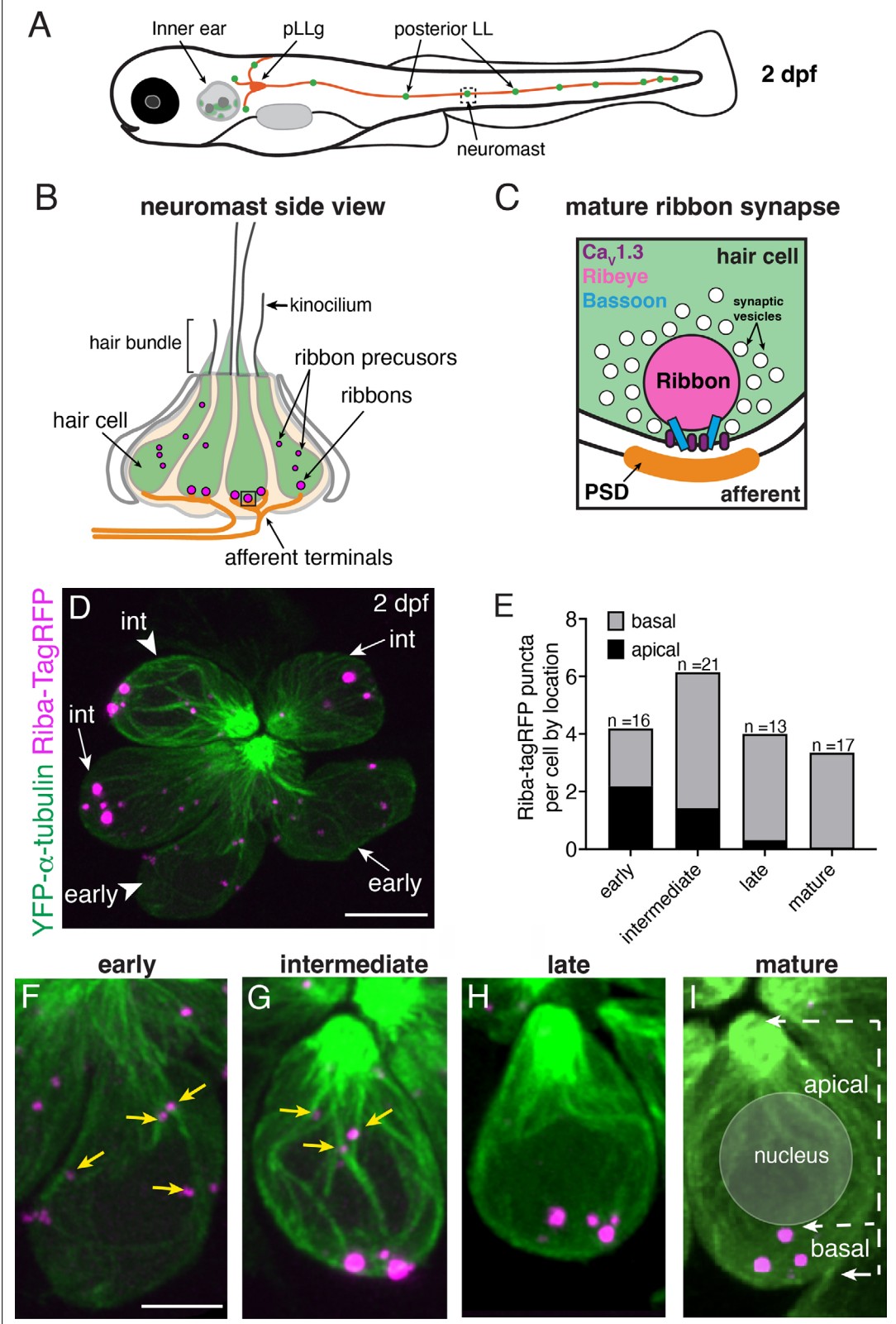

**Figure 1.** Ribbons associate with microtubules and change localization during development. (**A**) Schematic of a larval zebrafish at 2 days post fertilization (dpf) with the location of the posterior-lateral line (pLL) indicated relative to the inner ear. Neuromasts (green) in the pLL contain sensory hair cells that are innervated by afferent projections from the posterior lateral-line ganglion (pLLg, orange). (**B**) Schematic of a neuromast at 2 dpf, viewed from the side. At 2 dpf, the majority of hair cells (green) are developing. At the top of the cells, the mechanosensory hair bundle is composed

*Figure 1 continued on next page*

*Figure 1 continued*

of actin-based stereocilia and a tubulin-based kinocilium. A shorter kinocilium and an abundance of small ribbon precursors are indicative of an immature stage. (**C**) Schematic of a ribbon synapse when mature. The dense presynapse or ribbon is made primarily of Ribeye. Ribbons tether synaptic vesicles near Ca$_V$1.3 channels at the plasma membrane, across from the postsynaptic density (PSD). Bassoon acts to anchor ribbons at the presynaptic AZ. (**D**) Example image of a neuromast at 2 dpf, viewed from top down. The microtubule network and ribbons are marked with YFP-Tubulin and Riba-TagRFP, respectively. In this example of 6 developing hair cells, two early and four intermediate cells are present. The cell bodies of an early and intermediate cell from this example (arrowheads) are expanded in (**F** and **G**). (**E**) Plot shows the average number of ribbons per hair cell at each developmental stage. Cell stage is determined by the height of the kinocilium. After an increase in ribbon number with development, there is a decrease upon maturation. The number of apically localized Riba-TagRFP puncta is high at early and intermediate stages and is lower in late and mature hair cells. In contrast, the number of basally-localized Riba-TagRFP puncta is low at early stages and becomes higher by intermediate stages (n=16, 21, 13, 17 hair cells for early, intermediate, late, and mature stages, respectively). (**F–I**) Example images of hair cells expressing YFP-Tubulin and Riba-TagRFP at early, intermediate, late, and mature stages. At the early stage, Riba-TagRFP puncta are spread throughout the cell body and are smaller in size. At the intermediate stage, the number of Riba-TagRFP puncta becomes more basally enriched and are larger in size. In late and mature hair cells, all Riba-TagRFP puncta are at the base of the cell and are fewer in number compared to the intermediate stage. The arrows in (**I**) highlight the apical and basal regions of the cell used for quantification of Riba-TagRFP puncta location in (**E**). Yellow arrows in (**E**) and (**F**) indicate precursors associated with microtubules. Scale bars in D=5 µm and in F=2 µm.

The online version of this article includes the following figure supplement(s) for figure 1:

**Figure supplement 1.** Riba-TagRFP transgenic fish have normal cell numbers, synapses per hair cell, and ribbon areas.

**Figure supplement 2.** Ribbon number and apical-basal localization change during development.

of microtubules. We use these lines, along with high-resolution imaging to visualize the dynamics of ribbon formation. Using live imaging, we show that early in development, many small ribbon precursors are distributed throughout the cell. By later stages, fewer large ribbons remain and localize to the base of the cell. We show that microtubule networks in lateral-line hair cells are dynamic and grow plus ends towards the presynaptic AZ, the preferred direction for most kinesin motors (*Sweeney and Holzbaur, 2018*). Tracking analyses reveal the directed motion of ribbon precursors towards the presynaptic AZ, with ribbon precursors moving along and fusing on microtubules. Furthermore, we show that an intact microtubule network is critical for ribbon transport, fusion, and synapse formation. Overall, this foundational work provides insight into ribbon formation and the processes needed to reform ribbon synapses for the treatment of auditory and vestibular synaptopathies.

## Results

### Time course of ribbon formation in living zebrafish lateral-line hair cells

Fixed preparations in zebrafish and mice have outlined a conserved process that underlies ribbon or presynapse development in hair cells (*Michanski et al., 2019*; *Sheets et al., 2011*). To examine the time course underlying ribbon formation in vivo, we studied hair cells in the zebrafish pLL. These hair cells form 3–4 ribbon synapses in just 12–18 hr (*Dow et al., 2015*; *Kindt et al., 2012*). To image hair cells, ribbon precursors, and ribbons in vivo, we used a double transgenic line. One transgenic line labels microtubules and serves as a marker of hair cells (*myo6b:YFP-tubulin*). The other transgenic line reliably labels ribbons and smaller ribbon precursors (*myo6b:riba-TagRFP*) in hair cells (see example: *Figure 1D*). Although this latter transgene expresses Riba-TagRFP under a non-endogenous promoter, neither the tag nor the promoter ultimately impacts cell numbers, synapse counts, or ribbon size (*Figure 1—figure supplement 1A–E*).

For our initial analyses, we assessed the overall time course of ribbon formation in hair cells in the pLL when larvae were 2 dpf (*Figure 1A*). At 2 dpf, each pLL neuromast contains 4–8 developing hair cells at different developmental stages (*Figure 1B, D,F-I*). We staged hair cells based on the development of the apical, mechanosensory hair bundle. The hair bundle is composed of actin-based stereocilia and a tubulin-based kinocilium. We used the height of the kinocilium (see schematic in *Figure 1B*), the tallest part of the hair bundle, to estimate the developmental stage of hair cells as described previously (stage: hair bundle height; early: <1.5 µm, intermediate: 1.5–10 µm, late: 10–18 µm, mature: >18 µm *Zhang and Kindt, 2022*). Qualitatively, we observed that at early and intermediate stages, small Riba-TagRFP puncta were present throughout the cell body (see examples: *Figure 1D and F–G*). By late stages, or in newly matured hair cells, larger Riba-TagRFP puncta were restricted to the base of the cell (see example: *Figure 1H–I*). We quantified the total number of Riba-TagRFP puncta based on

developmental stage and found that the total number of puncta was high at early and intermediate stages but significantly decreased when hair cells were mature (*Figure 1E*, *Figure 1—figure supplement 2A*, n=7 neuromasts and 67 hair cells).

When taking these live images, we had no postsynaptic marker and were unable to differentiate between precursors and mature ribbons associated with a postsynaptic density. Therefore, we classified all apical Riba-TagRFP puncta above the nucleus as precursors, and all ribbons located beneath the nucleus at the cell base as more mature ribbons. Using this classification, we quantified the total number of Riba-TagRFP puncta at each developmental stage (*Figure 1E*, *Figure 1—figure supplement 2B-C*, n=7 neuromasts and 67 hair cells). We found that apical Riba-TagRFP precursors were only present at early and intermediate stages (*Figure 1E*, *Figure 1—figure supplement 2B*, and see yellow arrows in *Figure 1F–G*). In contrast, hair cells at late or mature stages contained more mature ribbons located at the cell base, below the nucleus (*Figure 1E, H-I*, *Figure 1—figure supplement 2C*). Overall, our live images examining Riba-TagRFP puncta show a similar, developmental process as observed in fixed preparations—the number of ribbons and precursors decrease and becomes basally localized to the presynaptic AZ as the hair cell develops.

## Microtubules grow plus ends toward the presynaptic active zone in hair cells

An important question in ribbon formation is how ribbons and precursors migrate to the presynaptic AZ. Work on mouse IHCs using electron microscopy (EM) and super-resolution microscopy found that developing ribbons and precursors often associate with microtubules (*Michanski et al., 2019*). Similar to what was observed in mice using EM, in living, pLL hair cells we observed ribbon precursors associated with microtubules (see yellow arrows in *Figure 1F–G*). Based on these association results, a microtubule network may function to transport ribbon precursors during development. To understand if ribbons could be transported along microtubules, we first examined the composition and polarity of the microtubule network in pLL hair cells.

We first examined the composition, or posttranslational modifications present in the microtubule network in pLL hair cells using immunohistochemistry. Using this approach, we labeled either acetylated- (modification found in mechanically stabilized microtubules) or tyrosinated- (modification found in destabilized microtubules) α-tubulin (*Janke and Magiera, 2020*). We performed our staining in a transgenic line that labels all microtubules (*myo6b:YFP-tubulin*). We found that in pLL hair cells, the microtubule network extends along the apical-basal axis of the cell and is highly acetylated; acetylated microtubules are more concentrated at the cell apex (*Figure 2—figure supplement 1A–C*). Outside of the kinocilium, we did not observe tyrosinated microtubules in pLL hair cells. Instead, tyrosinated microtubules were observed in the zebrafish skin, pLL nerve terminals, and the supporting cells that surround hair cells (*Figure 2—figure supplement 1D–F*). Overall, our immunostaining results indicate that in pLL hair cells, a considerable portion of the microtubule network is highly acetylated. Acetylation may provide a population of mechanically stable microtubules that could be used to transport ribbon precursors.

Although microtubule modifications are informative, these labels do not provide definitive information regarding microtubule growth or polarity. Knowing microtubule polarity is important, as many cargos are transported based on polarity. For example, most kinesin motor proteins transport cargo toward the more dynamic plus end of microtubules (*Sweeney and Holzbaur, 2018*). Therefore, to explore microtubule polarity in pLL hair cells, we created a transgenic line that expresses the plus-end marker of microtubules, EB3 (end-binding protein 3), fused with GFP (see example: *Figure 2A*, *myo6b:EB3-GFP*). Previous work in other cell types has shown that EB3-GFP can be used to visualize the plus end of growing microtubules (*Kawano et al., 2022*; *Stepanova et al., 2003*).

We used this transgenic line, along with confocal microscopy to visualize microtubule growth in pLL hair cells (z-stacks every 7 s for 20–30 min). By imaging EB3-GFP dynamics, we observed comet-like tracks that allowed us to visualize the plus end of growing microtubules (see *Figure 2—video 1*). In hair cells, microtubule organizing centers are located beneath a single microtubule-based kinocilium, the primary cilium in hair cells (*Lepelletier et al., 2013*). Consistent with studies in motile cilia, we observed foci of EB3-GFP at the tips of kinocilia that emanate away from the apex of the cell body (*Schrøder et al., 2011*; *Figure 2—figure supplement 2A–C*). This result suggests that similar to other cilia, the plus ends of kinocilia are towards their tips, away from the cell body. In addition, we

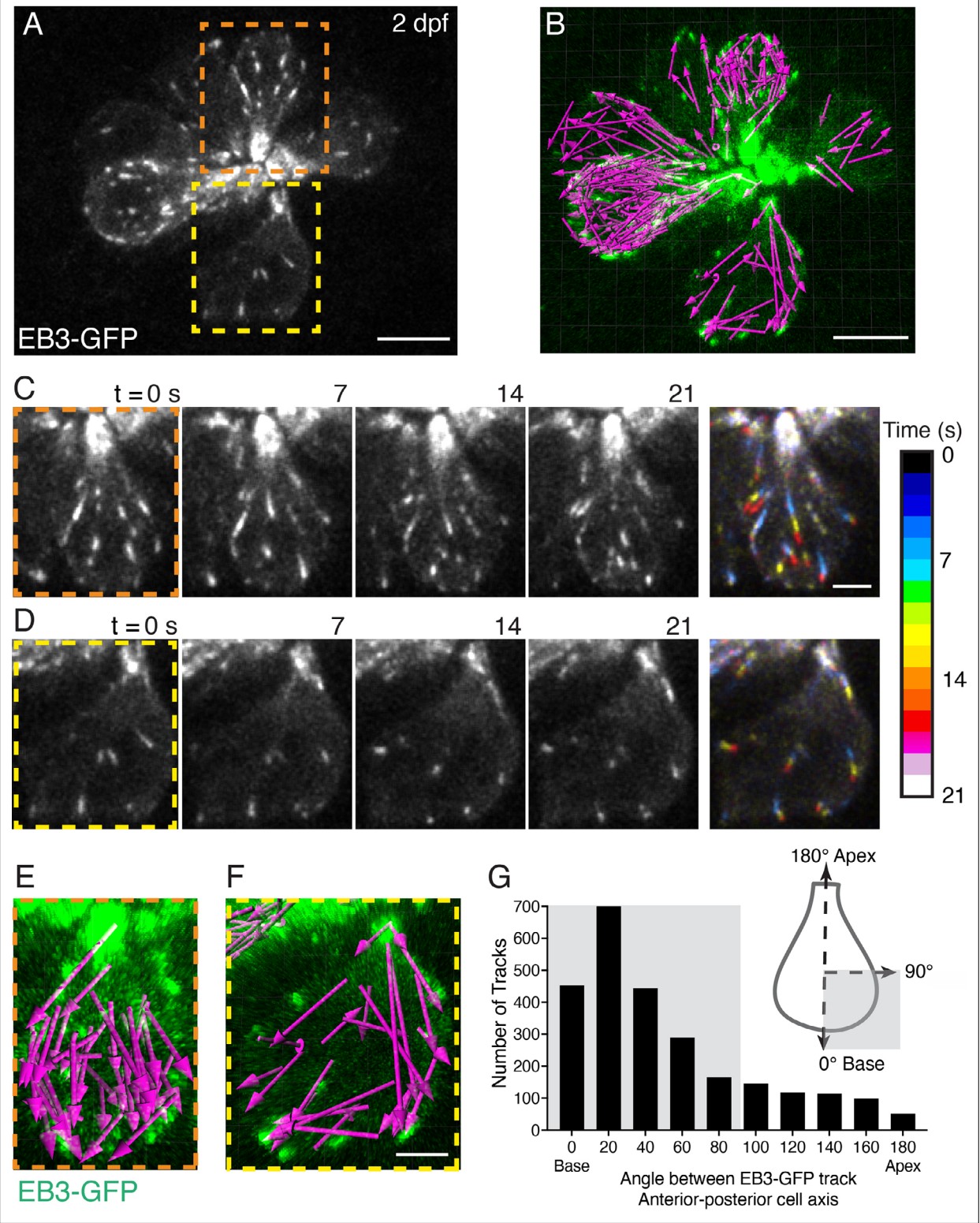

**Figure 2.** EB3-GFP tracks show plus ends of microtubules move to the cell base. (**A**) Example image of a neuromast at 2 dpf. The growing or plus ends of microtubules are marked with EB3-GFP. In this example, the apex of 8 developing cells is at the center of the image, and the base of each cell is at the periphery. Two example cells are outlined (dashed lines) and expanded in more detail in (**C–D**), and (**E–F**). (**B**) A 22-min timelapse was taken of the example in (**A**). All EB3 tracks, indicated by magenta arrows (tracked in Imaris) detected during the timelapse are shown. (**C–D**) Example time courses of EB3-GFP in hair cells over 21 s; the cell apex is towards the top of each image. In the final image, the four images for each example (0–21 s) were

*Figure 2 continued on next page*

*Figure 2 continued*

projected over time as a pseudocolor image represented by the colormap. The pseudocolor images show that many EB3-GFP tracks move to the cell base. (**E–F**) The magenta arrows in (**E**) and (**F**) show all the EB3-GFP tracks acquired in the example cells in (**C–D**) over the entire 22-min duration. Arrows indicate the direction of travel. (**G**) The schematic in (**G**) shows how EB3-GFP tracks were aligned to each hair cell. Tracks moving toward the apex have a track angle of 180°, while those moving to the base have an angle of 0°. This analysis revealed that the majority of EB3-GFP tracks (shaded domains) move toward the base of the cell (n=7 neuromasts, 2598 tracks). Scale bars in A-B=5 μm and in C-F=2 μm.

The online version of this article includes the following video and figure supplement(s) for figure 2:

**Figure supplement 1.** Microtubule modifications in lateral-line hair cells.

**Figure supplement 2.** EB3-GFP label reveals microtubule plus ends at kinocilial tips in hair cells.

**Figure 2—video 1.** EB3-GFP dynamics in developing lateral-line hair cells.

https://elifesciences.org/articles/98119/figures#fig2video1

observed EB3-GFP tracks within the soma of hair cells and found that the majority of tracks were directed away from the cell apex and towards the base of the cell (see EB3-GFP images taken from *Figure 2—video 1*, with tracks rendered into a pseudocolor image based on time, *Figure 2C–D*). To quantify the direction of EB3-GFP tracks, we used Imaris to perform 2D analyses to detect and create vectors of EB3-GFP tracks (see arrows; *Figure 2B, E and F*). We aligned these vectors along the 2D apical-basal axis of each cell. Here, vector movement towards the apex was represented as 180°, and movement to the base was represented at 0° (see schematic, *Figure 2G*). The movement of EB3-GFP vectors was quantified as the angle between 0 and 180°. From this analysis, we found that 80% of EB3-GFP vectors were directed towards the base in pLL hair cells (*Figure 2G*, 2069/2598 tracks <90°, n=7 neuromasts and 33 hair cells).

Overall, our immunostaining revealed that the soma of hair cells contains a population of microtubules stabilized by acetylation. Our in vivo imaging of EB3-GFP revealed that within this population there is extensive microtubule dynamics, and the plus ends of microtubules are primarily directed towards the cell base, and the presynaptic AZ.

## Ribbon precursors associate with and show directed motion along microtubules

Our analysis of microtubule dynamics using EB3-GFP indicates that the plus ends of microtubules point toward the cell base (*Figure 2*). Therefore, we hypothesized that these tracks of microtubules could be used by kinesin motor proteins to transport precursors from the cell apex to the base during development. To test this hypothesis, we used either Airyscan or Airyscan 2 confocal imaging to capture timelapses of ribbon and precursor movement for longer durations (Airyscan: ~3 μm z-stacks (15–20 slices) every 50–100 s for 30–70 min) or for shorter total durations with a faster capture rate (Airyscan 2: ~2–3.5 μm z-stacks (12–20 slices) every 3–20 s for 5–40 min). We focused our analysis on developing pLL hair cells at early and intermediate stages when ribbon precursors are abundant throughout the cell (*Figure 1D and F–G*). For this work, we used Riba-TagRFP to mark ribbons and precursors, and YFP-tubulin to label microtubules and provide cellular context.

From our timelapses, we observed that similar to our initial live analyses (*Figure 1F–G*), ribbon precursors associate with microtubules and are not found free-floating and untethered in the cell (*Figure 3—videos 1 and 2*). We observed that precursors exhibited three main movement behaviors related to microtubule association. First, we observed that the majority of precursors associated with microtubules remained stationary or confined (see examples: *Figure 3C* top two panels, *Figure 3D and E* asterisks, *Figure 3—video 3*). Second, we also observed rapid movement of precursors–during this movement, precursors appeared to be in close association with microtubules (see examples: *Figure 3C* bottom 4 panels, *Figure 3D and E* yellow arrowheads, *Figure 3—figure supplement 1A and B*, *Figure 3—videos 3, 4 and 5*). This movement along microtubules occurred bidirectionally, towards the cell apex and towards the base (*Figure 3C*, middle panels (arrows to base), bottom panels (arrows to apex), *Figure 3D-E*, *Figure 3—figure supplement 1A* and *Figure 3—videos 3 and 4* (to base), *Figure 3—figure supplement 1B* and *Figure 3—video 5* (to apex)). Third, during the timelapses, we often observed that precursors switched associations between neighboring microtubules (*Figure 3—figure supplement 1C–D*, *Figure 3—video 6*, 2.8 switching events per neuromast, n=10 neuromasts).

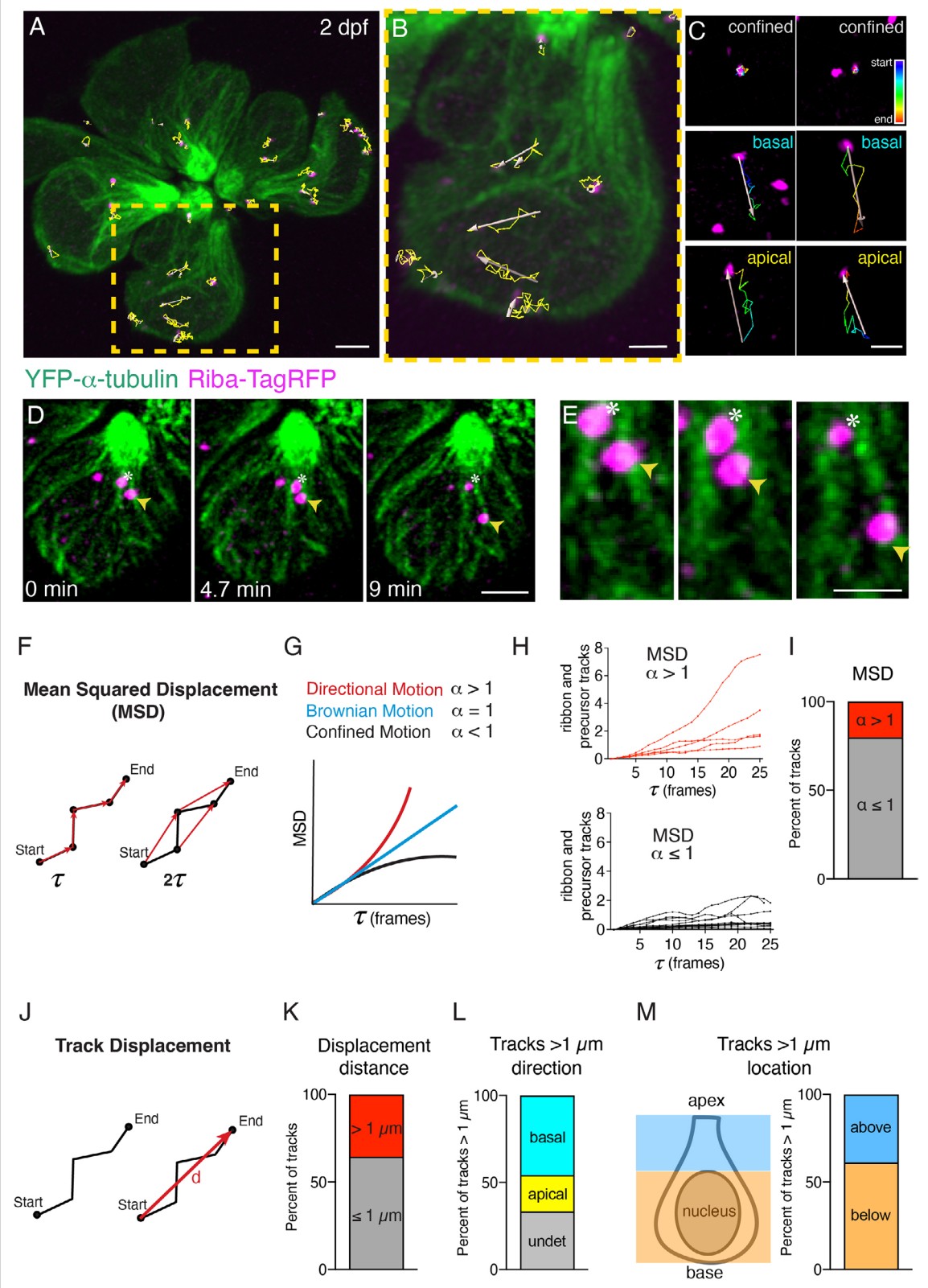

**Figure 3.** Ribbon precursors exhibit directional motion and confinement on microtubules. (**A –B**) To quantify motion, ribbons and ribbon precursors were tracked in hair cells at early and intermediate stages at 2 dpf (example). Shown are tracks (in yellow) from an entire neuromast (**A**) and a single hair cell (**B**), obtained using Imaris, during a 30-min timelapse acquired every 50 s (also see *Figure 3—video 1*). (**C**) Magnified view shows individual tracks over time and examples of confinement and motion toward the cell base and apex. Pseudo-colored tracks indicate the timecourse of movement

*Figure 3 continued on next page*

*Figure 3 continued*

(blue start, red end). (**D**) Example of confined motion and directed motion on microtubules in a single hair cell. Images were obtained during a 9-min timelapse acquired every 20 s. The ribbon labeled by the asterisks remains confined, while the ribbon labeled with the yellow arrowheads moves along a microtubule, towards the cell base, over time (also see *Figure 3—video 3*). (**E**) A magnified image of the example shown in (**D**). (**F**) Mean squared displacement (MSD) vs time step was used to measure movement behaviors. Shown in red are the first- and second-time steps. The results are plotted in the form of MSD vs time step ($\tau$) plots and the exponent ($\alpha$) of the plots can be used to distinguish between the different types of motion observed: confined ($\alpha < 1$), directional ($\alpha > 1$), or Brownian motion ($\alpha = 1$). (**H**) Example MSD plots of individual ribbon tracks from 2 control neuromasts (15 tracks MSD < 1 (black), five tracks MSD > 1 (red)). (**I**) The bar graph shows the percent of MSD tracks displaying confined ($\alpha < 1$, 79.8%, gray), and directional motion ($\alpha > 1$, 20.2%, red). (**J**) Track displacement vs time was used to measure movement behaviors with track > 1 µm indicative of directed motion. (**K**) The bar graph shows the percent of tracks with distances > 1 µm (35.6% red) and those with distances ≤ 1 µm (65.4%, gray). (**L**) The bar graph shows the percent of tracks with distances > 1 µm based on track direction (45.8% to the cell base, cyan; 20.8% to the cell apex, yellow; 33.3% undetermined direction, gray). (**M**) The bar graph shows the percent of tracks with distances > 1 µm based on location in the cell, above or below the nucleus (38.9% above the nucleus, blue; 61.1% below the nucleus, orange). In (**K –M**) n = 10 neuromasts, 40 hair cells, and 203 tracks. Scale bar in A, D=2 µm and B, C, E=1 µm.

The online version of this article includes the following video and figure supplement(s) for figure 3:

**Figure supplement 1.** Ribbons move directionally on microtubules and can move between microtubule filaments.

**Figure 3—video 1.** Tracking precursors and ribbons in 3D using Imaris.

https://elifesciences.org/articles/98119/figures#fig3video1

**Figure 3—video 2.** Tracking precursors and ribbons in 3D using Imaris.

https://elifesciences.org/articles/98119/figures#fig3video2

**Figure 3—video 3.** Directional motion to cell base and stationary precursors on microtubule.

https://elifesciences.org/articles/98119/figures#fig3video3

**Figure 3—video 4.** Directional motion of precursor along a microtubule to the cell base.

https://elifesciences.org/articles/98119/figures#fig3video4

**Figure 3—video 5.** Directional motion of precursor along microtubule to cell apex.

https://elifesciences.org/articles/98119/figures#fig3video5

**Figure 3—video 6.** Precursor switching between microtubules.

https://elifesciences.org/articles/98119/figures#fig3video6

To quantify ribbon and precursor movement, we used Imaris to obtain x, y, z coordinates for each ribbon during our timelapses (longer 30–70 min acquisitions) (see examples: *Figure 3A–C*, *Figure 3— videos 1 and 2*). This analysis yielded tracks or trajectories for all precursors and ribbons. We then performed a mean-squared displacement (MSD) analysis on ribbon tracks to classify the type of motion we observed. In this analysis, the exponent ($\alpha$) of MSD vs time is obtained by curve fitting. A value of $\alpha > 1$ indicates directional motion with velocity, $\alpha = 1$ indicates Brownian motion, and $\alpha < 1$ is representative of confined motion or subdiffusion (*Figure 3F and G*; *Sikora et al., 2017*). This method allows us to determine what type of motion each track is exhibiting. Our results show that in developing hair cells, ribbon tracks exhibit directional as well as confined motion (*Figure 3H*, for example MSD tracks with $\alpha > 1$ (red tracks, top panel) and $\alpha < 1$ (black tracks, bottom panel) from two neuromasts). Upon quantification, 20.2% of ribbon tracks show $\alpha > 1$, indicative of directional motion, but the majority of ribbon tracks (79.8%) show $\alpha < 1$, indicating confinement on microtubules (*Figure 3I*, n=10 neuromasts, 40 hair cells, and 203 tracks).

To provide a more comprehensive analysis of precursor movement, we also examined displacement distance (*Figure 3J*). Here, as an additional measure of directed motion, we calculated the percent of tracks with a cumulative displacement >1 µm. We found that 35.6% of tracks had a displacement >1 µm (*Figure 3K*; n=10 neuromasts, 40 hair cells, and 203 tracks). Of the tracks with displacement >1 µm, the majority of ribbon tracks (45.8%) moved to the cell base, but we also found a subset of ribbon tracks (20.8%) that moved apically (33.4% moved in an undetermined direction) (*Figure 3L*). This apical movement is consistent with a subpopulation of microtubules showing plus-end mediated growth apically (20.4% of EB3-GFP tracks). In addition, we examined the location of precursors within the cell that exhibited displacements >1 µm. We found that 38.9% of these tracks were located above the nucleus, while 61.1% were located below the nucleus (*Figure 3M*). Overall, our timelapse imaging demonstrated that ribbons and precursors displayed three main types of movement on microtubules: confinement, movement along microtubules, and switching between microtubules. Furthermore,

our tracking analyses indicate that while the majority of precursors are confined on microtubules, a subpopulation of ribbons and precursors exhibits directional motion.

## Long-term manipulation of microtubules impacts ribbon formation

Our timelapse imaging revealed that precursors and ribbons can move along directionally microtubules. To assess the importance of the microtubule network in synapse formation, we used pharmacology to destabilize or stabilize the microtubule network in pLL hair cells using nocodazole or taxol, respectively. We incubated larvae at 2 dpf for 16 hr (56–72 hours post fertilization (hpf)), a time window that encompasses a large portion of synapse formation in developing hair cells. After these pharmacological treatments, we fixed and immunostained larvae to label acetylated-α-tubulin, to monitor changes to the microtubule network. In addition, we co-labeled with Ribeyeb to label ribbons and precursors and pan-Maguk to label postsynapses.

After a 16 hr treatment with 250 nM nocodazole, we observed a decrease in acetylated-α-tubulin label (qualitative examples: *Figure 4A, C*, *Figure 4—figure supplement 1A-B*). Quantification revealed significantly less mean acetylated-α-tubulin label in hair cells after nocodazole treatment (*Figure 4—figure supplement 1D*). Less acetylated-α-tubulin label indicates that our nocodazole treatment successfully destabilized microtubules. We also examined the number of hair cells per neuromast and observed that after nocodazole treatment, there were significantly fewer hair cells compared to controls. This indicates that either nocodazole is slightly toxic or interferes with cell division. Both situations have been observed previously during nocodazole treatments in other systems (*Gupta, 1985*; *Zieve et al., 1980*). We next examined the Ribeyeb label in hair cells to assess precursors and ribbons. We found that after nocodazole treatment, the total number of Ribeyeb puncta (apical and basal) per hair cell was significantly higher compared to controls (*Figure 4G*). We also observed that the average of individual Ribeyeb puncta (from 2D max-projected images) was significantly reduced compared to controls (*Figure 4H*). Furthermore, the relative frequency of individual Ribeyeb puncta with smaller areas was higher in nocodazole-treated hair cells compared to controls (*Figure 4I*). We also examined the number of complete synapses (Ribeyeb-Maguk paired puncta) per hair cell after nocodazole treatment. We found that there were significantly fewer complete synapses per hair cell after a 16 hr nocodazole treatment (*Figure 4F*). Our long-term nocodazole treatment indicates that microtubule destabilization led to an increase in ribbons and precursors while reducing the number of synapses per cell.

We performed a similar analysis on hair cells after a 16 hr treatment with 25 µM taxol. After treatment, we observed more acetylated-α-tubulin label, indicating that our taxol treatment successfully stabilized microtubules (qualitative examples: *Figure 4—figure supplement 1A and C*, *Figure 4—figure supplement 2A and C*). Quantification revealed an overall increase in mean acetylated-α-tubulin label in hair cells after taxol treatment, but this increase did not reach significance (*Figure 4—figure supplement 1E*). Unlike nocodazole treatment, taxol did not significantly impact the number of hair cells, the average area of Ribeyeb puncta, or the number of ribbons and precursors (apical and basal) per hair cell compared to controls (*Figure 4—figure supplement 2E, G–I*). Interestingly, we observed slightly more complete synapses per hair cell after a 16 hr taxol treatment (*Figure 4—figure supplement 2F*). Overall, our long-term taxol treatment indicates that microtubule stabilization does not dramatically impact synapse formation in pLL hair cells.

Together, this pharmacology study revealed that long-term destabilization and stabilization of microtubules during development can impact ribbon formation in pLL hair cells. Microtubule destabilization had the most dramatic effect leading to more ribbons and precursors, and the formation of fewer complete synapses.

## *Kif1aa* mutants have fewer synapses and more ribbon precursors

For cargo (such as ribbons and precursors) to be transported along microtubules, molecular motor proteins are required. Kinesin motor proteins transport cargo along microtubules towards the growing, plus end. Our work demonstrated that in hair cells of the pLL, the plus end of microtubules grow from the apex to the base of the cell (*Figure 2*). Single-cell RNA sequencing (scRNAseq) has revealed that *kif1aa*, a zebrafish orthologue of mammalian Kif1a, a plus-end directed kinesin motor protein, is highly expressed in pLL hair cells (*Lush et al., 2019*). In zebrafish, there are two orthologues of mammalian *Kif1a*, *kif1aa* and *kif1ab*. ScRNA-seq in zebrafish has demonstrated widespread co-expression of

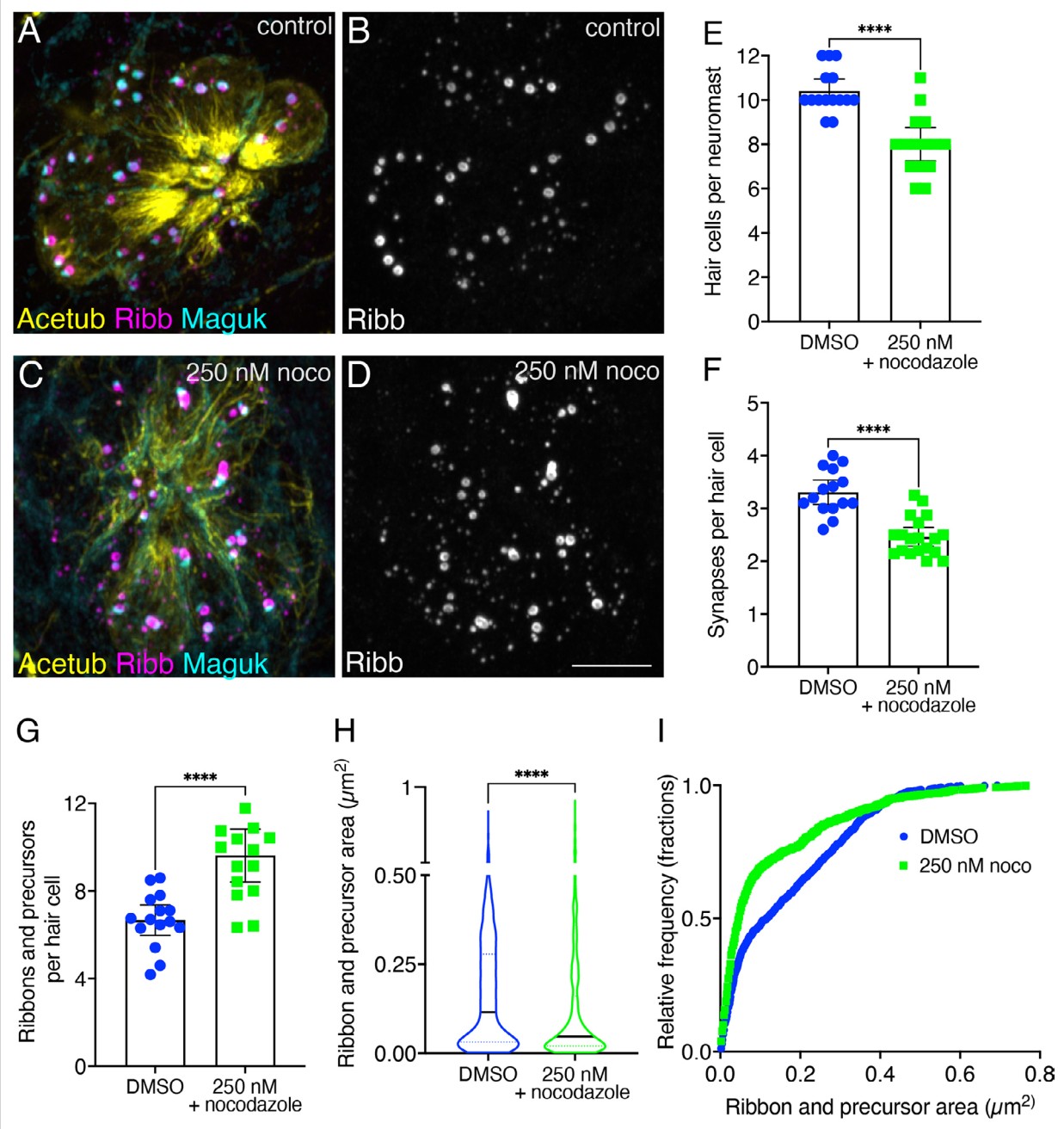

**Figure 4.** Overnight microtubule destabilization increases ribbon numbers, and decreases synapse counts and ribbon areas. (**A–D**) Example immunostain of a neuromast at 3 dpf after an overnight treatment with 250 nM nocodazole (**C–D**) or DMSO (**A–B**). Acetylated-α-tubulin (Acetub) labels microtubules, Ribeyeb (Ribb) labels precursors and ribbons, and Maguk labels postsynapses. (**E–G**) After an overnight treatment with 250 nM nocodazole, there are significantly fewer hair cells per neuromast (**E**), p<0.0001, fewer complete synapses per cell (**F**), p<0.0001, and more ribbons and precursors per cell (**G**), p<0.0001 compared to controls (n=15 neuromasts for control and 250 nM nocodazole treatments). (**H–I**) After an overnight treatment with 250 nM nocodazole the average area of Ribb puncta was significantly lower compared to controls (**H**), p<0.0001 (n=1008 and 1135 Ribb puncta for control and 250 nM nocodazole treatments). In (**I**), the relative frequency of all the areas of Ribb puncta are plotted in nocodazole treatment and controls. For comparisons, an unpaired t-test was used in (**E–G**), and a Mann-Whitney test was used in (**H**). Error bars represent SEM in E-G. In H the median and first and third quartiles are shown. Scale bar in D=5 μm.

The online version of this article includes the following video and figure supplement(s) for figure 4:

**Figure supplement 1.** Nocodazole and taxol treatment impact microtubules in lateral-line hair cells.

**Figure supplement 2.** Overnight microtubule stabilization slightly increases synapse counts.

*Figure 4 continued on next page*

*Figure 4 continued*

**Figure 4—video 1.** Microtubule dynamics in hair cells change upon treatment with nocodazole and taxol.
https://elifesciences.org/articles/98119/figures#fig4video1

---

*kif1ab* and *kif1aa* mRNA in the nervous system. Additionally, both scRNA-seq and fluorescent in situ hybridization have revealed that pLL hair cells exclusively express *kif1aa* mRNA (*David et al., 2024*; *Lush et al., 2019*; *Sur et al., 2023*). Therefore, we tested whether Kif1aa could be the kinesin motor that transports ribbons and precursors to the cell base during development.

To test for the role of Kif1aa in pLL hair cells, we created a CRISPR-Cas9 mutant (*Varshney et al., 2016*). Our *kif1aa* mutant has a stop codon in the motor domain and is predicted to be a null mutation (*Figure 5—figure supplement 1A–B*). Recent work in our lab using this mutant has shown that Kif1aa is responsible for enriching glutamate-filled vesicles at the base of hair cells. In addition, this work demonstrated that loss of Kif1aa results in functional defects in mature hair cells, including a reduction in evoked post-synaptic calcium responses (*David et al., 2024*). We hypothesized that Kif1aa may also be playing an earlier role in ribbon formation.

For our initial analysis of *kif1aa* mutants, we co-labeled hair cells at 3 dpf with Ribeyeb to label ribbons and precursors and pan-Maguk to label postsynapses (see examples: *Figure 5A–D*). This is a similar endpoint used to examine synapse formation after our long-term nocodazole and taxol treatments (*Figure 4*, *Figure 4—figure supplement 2*). We found that at 3 dpf *kif1aa* mutants had a similar number of hair cells per neuromast (*Figure 5E*). Despite a similar number of hair cells, we found that there were significantly fewer complete synapses per hair cell in *kif1aa* mutants compared to controls (*Figure 5F*). In addition, we found that there were significantly more ribbons and precursors (apical and basal) in *kif1aa* mutants compared to controls (*Figure 5G–I*). As described in the previous section, we also observed fewer complete synapses, and more ribbons and precursors after a 16 hr nocodazole treatment (*Figure 4F–G*). Together, this suggests that both intact microtubules and Kif1aa are required for normal synapse formation in pLL hair cells.

## Short-term disruption of microtubules, but not loss of Kif1aa, impacts ribbon formation

Our initial experiments suggest that microtubule networks and Kif1aa are important for proper synapse formation in pLL hair cells. However, the actual changes in precursors and ribbons within developing hair cells were unclear. Therefore, we examined changes in ribbons and precursors in living pLL hair cells over 3–4 hr of development. For our analysis, we used transgenic lines expressing YFP-tubulin to monitor microtubules and Riba-TagRFP to monitor precursors and ribbons in vivo. We examined precursors and ribbons after nocodazole or taxol treatment, or after knockdown of Kif1aa during this developmental window.

We verified the effectiveness of our in vivo pharmacological treatments using either 500 nM nocodazole or 25 µM taxol by imaging microtubule dynamics in pLL hair cells (*myo6b:YFP-tubulin*). After a 30-min pharmacological treatment, we used Airyscan confocal microscopy to acquire timelapses of YFP-tubulin (3 µm z-stacks, every 50–100 s for 30–70 min, *Figure 4—video 1*). Compared to controls, 500 nM nocodazole destabilized microtubules (presence of depolymerized YFP-tubulin in the cytosol, see arrows in *Figure 4—figure supplement 1F–G*) and 25 µM taxol dramatically stabilized microtubules (indicated by long, rigid microtubules, see arrowheads in *Figure 4—figure supplement 1F and H*) in pLL hair cells. We did still observe a subset of apical microtubules after nocodazole treatment, indicating that this population is particularly stable (see asterisks in *Figure 4—figure supplement 1F–H*).

After verifying our in vivo pharmacological pharmacology treatments, we acquired Airyscan confocal images of developing hair cells at 2 dpf that express YFP-tubulin and Riba-TagRFP. Then after a 3–4 hr treatment in either 500 nM nocodazole or 25 µM taxol, we reimaged the same hair cells. Consistent with our previous results (*Figure 4*, *Figure 4—figure supplement 1*) nocodazole and taxol treatments destabilized or stabilized microtubules, respectively (qualitative examples: *Figure 6A–F*). In each developing cell, we quantified the total number of Riba-TagRFP puncta (apical and basal) before and after each treatment. In our control samples, we observed on average no change in the number of Riba-TagRFP puncta per cell (*Figure 6G*). Interestingly, we observed that nocodazole treatment led to a significant increase in the total number of Riba-TagRFP puncta after 3–4 hr (*Figure 6G*). This

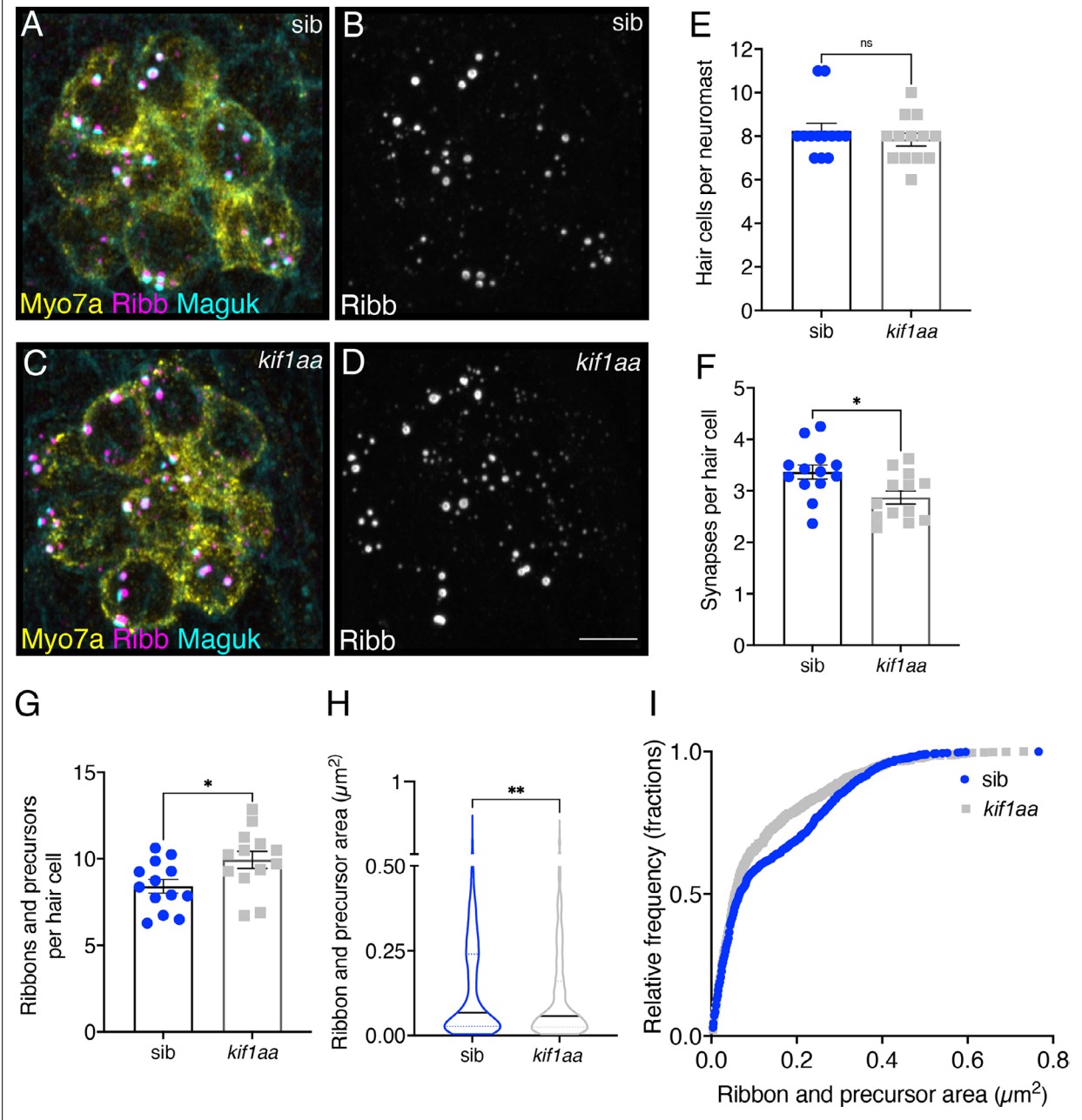

**Figure 5.** Loss of Kif1aa increases precursor numbers and decreases synapse counts. (**A–D**) Example immunostain of neuromasts at 3 dpf in *kif1aa* germline mutants (**C–D**) or sibling control (**A–B**). Myosin7a labels hair cells, Ribeyeb (Ribb) labels precursors and ribbons, and Maguk labels postsynapses. (**E–G**) In *kif1aa* mutants, there is no change in the number of hair cells per neuromast (**E**), p=0.418, but there are fewer complete synapses per cell (**F**), p=0.014, and more ribbons and precursors per cell (**G**), p=0.024 compared to sibling controls (n=13 neuromasts for control and *kif1aa* mutants). (**H–I**) In *kif1aa* germline mutants, the average area of Ribb puncta was significantly lower compared to sibling controls (**H**), p=0.005 (n=896 and 1008 Ribb puncta for control and *kif1aa* mutants). In (**I**), the relative frequency of all the areas of all Ribb puncta are plotted in *kif1aa* mutants and sibling controls. For comparisons, an unpaired t-test was used in (**E–G**), and a Mann-Whitney test was used in (**H**). Error bars represent SEM in E-G. In H the median and first and third quartiles are shown. Scale bar in D=5 µm.

The online version of this article includes the following figure supplement(s) for figure 5:

**Figure supplement 1.** Kif1aa protein and exon 6 lesions.

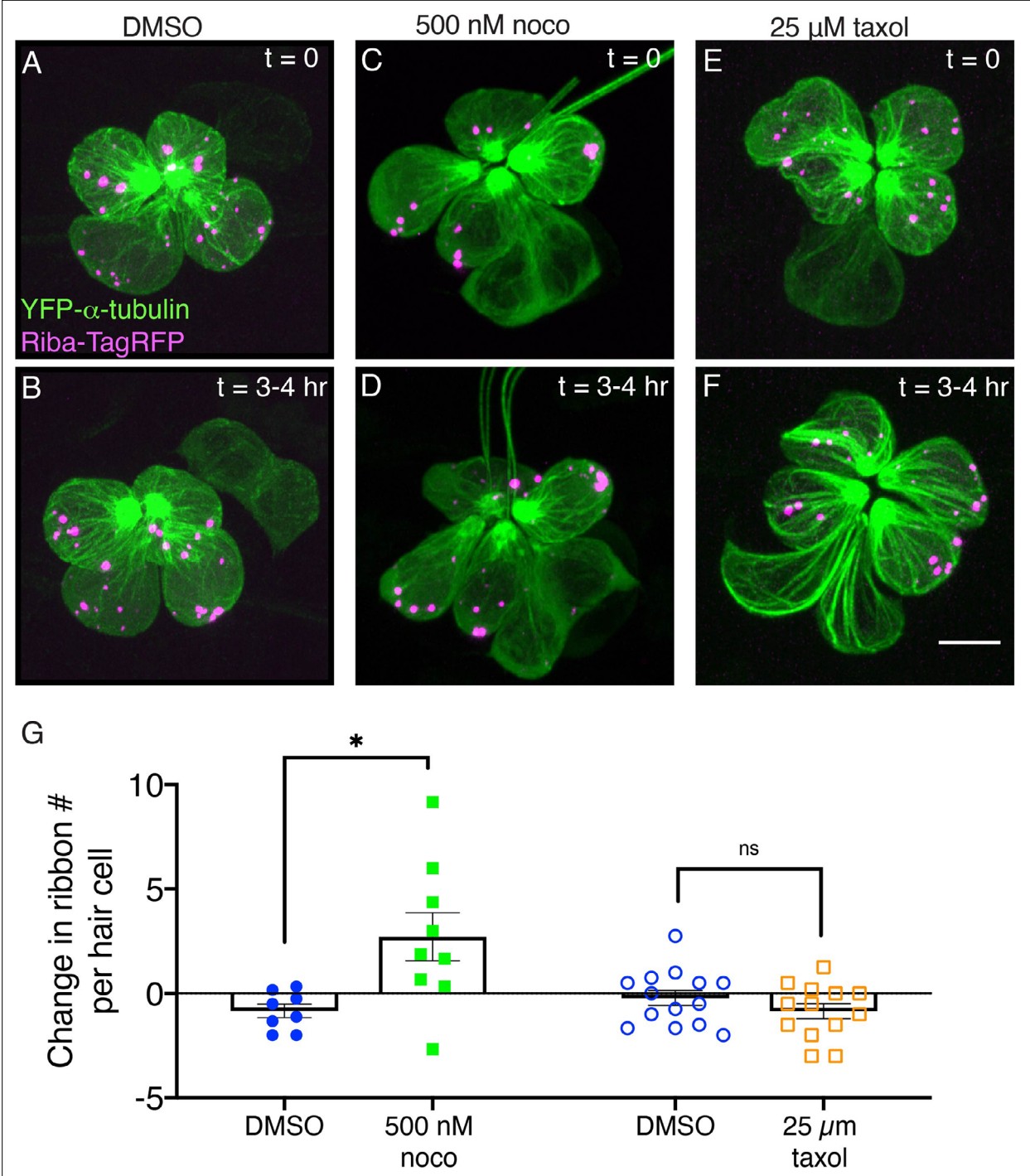

**Figure 6.** Microtubule destabilization increases ribbon numbers in vivo. (**A, C, E**) Example images of neuromasts at 2 dpf. The microtubule network and ribbons are marked with YFP-tubulin and Riba-TagRFP, respectively. Neuromasts were imaged immediately after a 30-min treatment with DMSO (control) (**A**), 500 nM nocodazole (**C**) or 25 μM taxol (**E**), (t=0). (**B, D, F**) The same neuromasts in A, C, E were reimaged after an additional 3–4 hr of treatment. (**G**) Quantification reveals that after 3–4 hr nocodazole treatment, there are more Riba-TagRFP puncta per hair cell compared to controls (n=9 and 8 neuromasts for nocodazole and DMSO, p=0.013). In contrast, after a 3–4 hr taxol treatment, there was no significant change in the number of Riba-TagRFP puncta per hair cell compared to controls (n=13 and 14 neuromasts for taxol and DMSO, p=0.256). An unpaired t-test was used for comparisons in (**G**). Error bars represent SEM. Scale bar in (**F**)=5 μm.

The online version of this article includes the following figure supplement(s) for figure 6:

**Figure supplement 1.** *kif1aa* crispant verification via genotyping via fluorescent fragment analysis.

**Figure supplement 2.** Loss of Kif1aa does not impact ribbon numbers over 3–4 hr.

result is similar to our overnight nocodazole experiments in fixed samples, where we also observed an increase in the number of ribbons and precursors per hair cell. In contrast to our 3–4 hr nocodazole treatment, similar to controls, taxol treatment did not alter the total number of Riba-TagRFP puncta over 3–4 hr (*Figure 6G*). Overall, our overnight and 3–4 hr pharmacology experiments demonstrate that microtubule destabilization has a more significant impact on ribbon numbers compared to microtubule stabilization.

Next, we used a similar approach to examine the number of Riba-TagRFP puncta in *kif1aa* mutants over a 3–4 hr time window. For this analysis, we examined puncta in *kif1aa* F0 crispants. These mutants are derived from the injection of 2 *kif1aa* guide RNAs (gRNAs) and Cas9 protein. This approach has shown to be an extremely effective way to assay gene function in any genetic background (*Hoshijima et al., 2019*). Using this approach, we were able to robustly disrupt the *kif1aa* locus (*Figure 6—figure supplement 1*). We found that compared to uninjected controls, in *kif1aa* F0 crispants there was no change in the total number of Riba-TagRFP puncta per cell over 3–4 hr (*Figure 6—figure supplement 2A–F*). This result is slightly different than our analysis of *kif1aa* mutants at 3 dpf where we observed significantly more ribbons and precursors per cell compared to controls (*Figure 5G*). Overall, live imaging over a 3–4 hr time window indicates that a loss of microtubule stability, but not loss of Kif1aa, results in an increase in Riba-TagRFP puncta within developing pLL hair cells.

## Ribbons and precursors require microtubules, but not Kif1aa for directed motion

Our tracking analyses revealed that ribbons and precursors show directed motion on microtubules. Based on these results, we asked what happens to this movement if we disrupt microtubules or the kinesin motor Kif1aa. To test whether microtubules are required for directional ribbon and precursor motion, we used the drugs nocodazole and taxol to alter microtubule dynamics. We treated the fish with these drugs for 30 min and recorded timelapses of ribbon and precursor motion. This short treatment allowed us to observe ribbon motion relatively soon after microtubule disruption and minimize any cytotoxic effects of the compounds. Then we performed MSD and track displacement analyses on ribbon movement to determine if directional motion was impaired (*Figure 7A–B*).

Using this approach, we found that after destabilizing microtubules by treatment with 500 nM nocodazole for 30 min, the proportion of tracks with $\alpha>1$ was significantly reduced compared to controls (*Figure 7C*). In addition, we also found that the proportion of longer tracks with a cumulative displacement >1 µm was also reduced after nocodazole treatment compared to controls (*Figure 7F*). This analysis indicates that an intact microtubule network is needed for proper directional ribbon motion and longer displacements. In contrast, after stabilizing microtubules by treatment with 25 µM taxol, we observed no effect on the proportion of tracks with displacement >1 µm or the proportion of tracks with $\alpha>1$, compared to controls (*Figure 7D and G*). Interestingly, when we examined the distribution of $\alpha$ values, we observed that taxol treatment shifted the overall distribution towards higher $\alpha$ values (*Figure 7—figure supplement 1A*). In addition, when we plotted only tracks with directional motion ($\alpha > 1$), we found significantly higher $\alpha$ values in hair cells treated with taxol compared to controls (*Figure 7—figure supplement 1B*). This indicates that in taxol-treated hair cells, where the microtubule network is stabilized, ribbons with directional motion have higher velocities.

Next, we used a similar approach to examine the tracks of ribbon and precursor motion in *kif1aa* mutants. For this analysis, we tracked ribbons and precursors in *kif1aa* F0 crispants. We observed that the proportion of tracks with displacement >1 µm was not significantly different between *kif1aa* F0 crispants and controls (*Figure 7H*). Similarly, the proportion of tracks with $\alpha>1$, was also not significantly different from the controls (*Figure 7E*). Overall, these tracking analyses show that in developing pLL hair cells, the kinesin motor Kif1aa is not essential for the directional motion of ribbons and precursors. In contrast, an unperturbed microtubule network is essential for directed motion.

## Ribbon precursors fuse on microtubules

Our analyses demonstrate that the movement of ribbon precursors along microtubules is important for synapse formation. But in addition to movement, during development, it has been proposed that these small precursors come together or fuse to form larger, more mature ribbons (*Michanski et al., 2019*). This is consistent with our work, where we see a reduction in ribbon and precursor numbers alongside development (*Figure 1E*). Furthermore, we see an increase in ribbon and precursor number

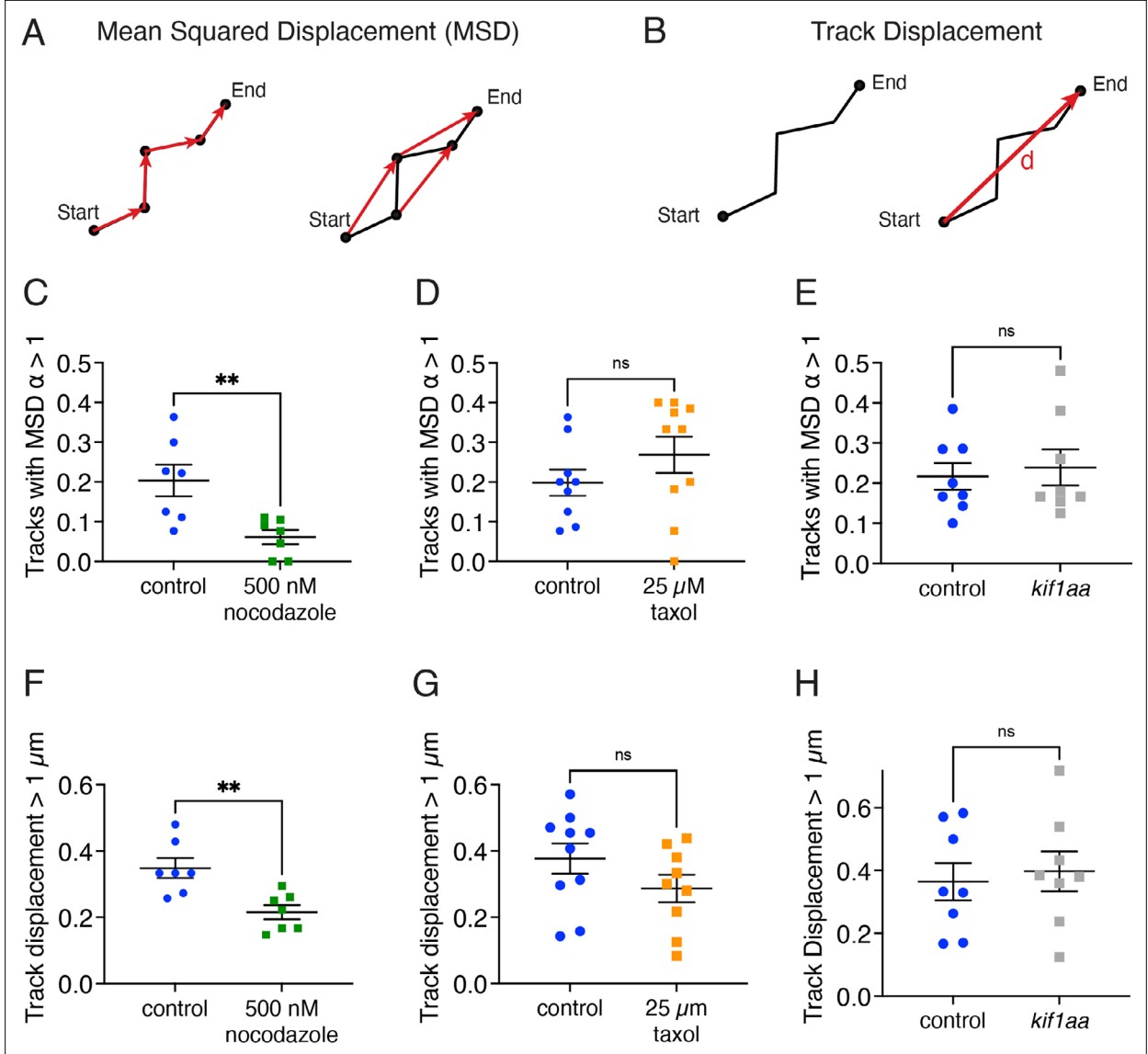

**Figure 7.** Ribbon precursors require intact microtubules, but not Kif1aa for directional motion. (**A**) Mean squared displacement (MSD) vs time was used to quantify the proportion of ribbon and precursor tracks with a velocity and α>1, a behavior indicative of directionally moving tracks. Shown are the first- and second-time steps. (**B**) To further quantify directional motion, the proportion of tracks with large displacements >1 µm was quantified. Track displacement was measured as the distance between the start and end point of the track. (**C**) Hair cells treated with 500 mM nocodazole have fewer directional tracks (α>1) compared to controls (p=0.007). (**D**) In hair cells treated with 25 µM taxol, there are not significantly fewer directional tracks (α>1) compared to controls (p=0.24). (**E**) In hair cells lacking Kif1aa, there are not significantly fewer directional tracks (α>1) compared to controls (p=0.70). (**F**) Hair cells treated with 500 nM nocodazole have fewer ribbons with track displacements >1 µm compared to control (p=0.004). (**G**) There is no change in track displacement >1 µm in hair cells treated with 25 µM taxol (p=0.17). (**H**) There is no change in track displacement >1 µm in hair cells lacking Kif1aa (p=0.71). N=7 neuromasts for DMSO control and nocodazole for (**C**) and (**F**); n=9 and 10 neuromasts for DMSO control and taxol for (**D**) and (**G**); n=8 neuromasts for control and *kif1aa* for (**E**) and (**H**). An unpaired t-test was used in (**C–H**). Error bars represent SEM.

The online version of this article includes the following figure supplement(s) for figure 7:

**Figure supplement 1.** A more stable microtubule network results in directional ribbon tracks with higher mean squared displacement (MSD) α values.

when microtubules are destabilized (*Figure 4*), indicating an intact microtubule network may also be important for ribbon fusion.

Consistent with this idea, in our timelapses, we observed that ribbons and precursors undergo fusion on microtubules. These fusion events occurred between smaller precursors at the cell apex as well as between larger precursors near the cell base (see examples: *Figure 8A-B*, *Figure 8—figure supplement 1A-B*, *Figure 8—videos 1 and 2, 3*). We classified an event as a fusion once the ribbons

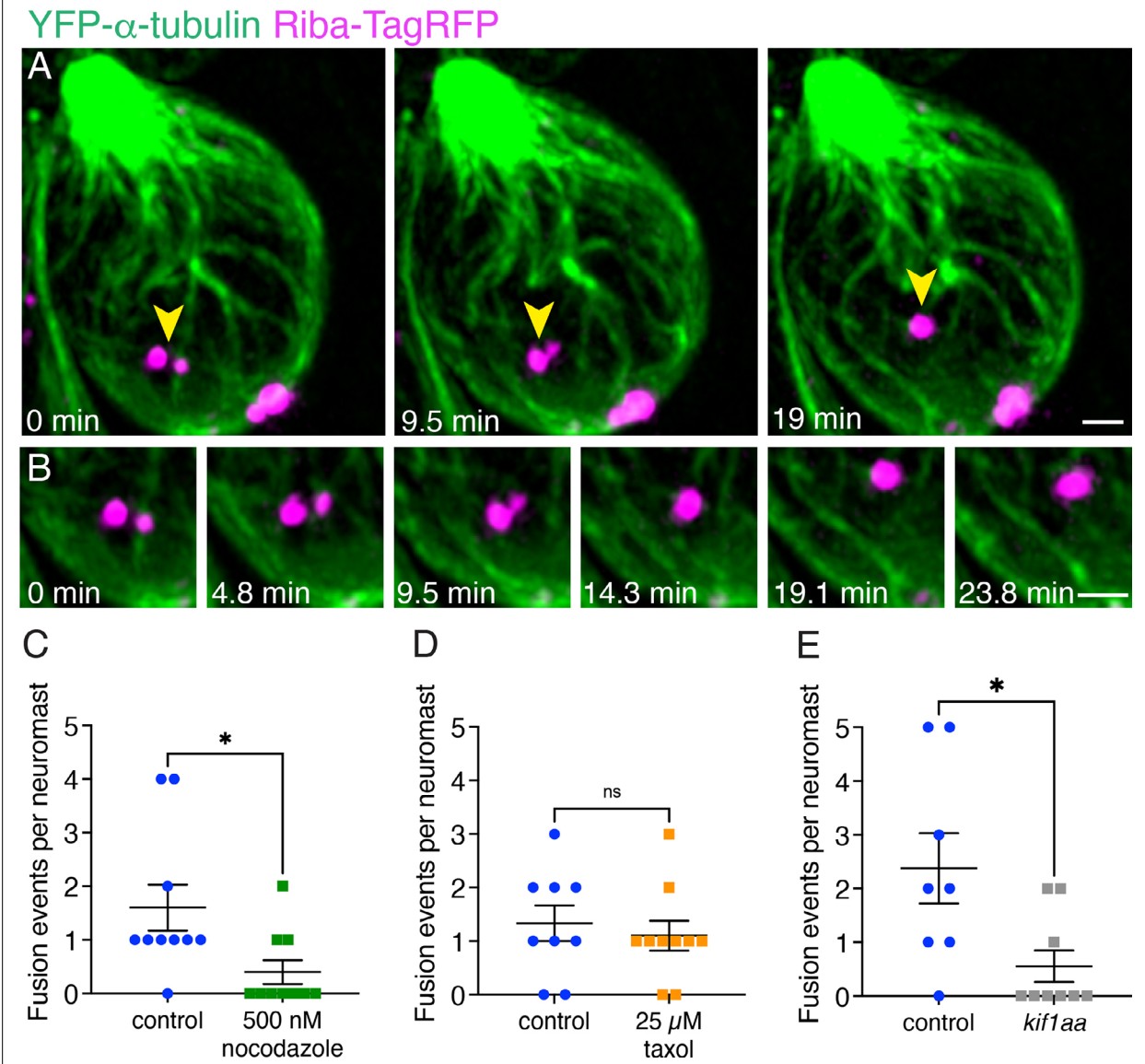

**Figure 8.** Microtubules and Kif1aa are required for fusion events. (**A**) An example of two ribbon precursors undergoing fusion on microtubules (yellow arrowheads). (**B**) A zoomed-in montage of the example from (**A**) is shown where the association of each precursor with microtubules can be seen during the process of fusion (also see *Figure 8—video 1*). (**C–D**) Destabilizing the microtubules with nocodazole treatment reduces the number of fusion events observed in timelapses (p=0.023). Taxol treatment has no effect (p=0.60). (**E**) Loss of Kif1aa significantly reduces the number of fusion events observed in timelapses compared to control (p=0.018). N=10 neuromasts for DMSO control and nocodazole for (**C**); n=9 and 10 neuromasts for DMSO control and taxol for (**D**); n=8 and 9 neuromasts for control and *kif1aa* for (**E**). An unpaired t-test was used for comparisons in (**C–E**). Error bars represent SEM. Scale bars in (**A**) and B=1 μm.

The online version of this article includes the following video and figure supplement(s) for figure 8:

**Figure supplement 1.** Ribbon precursor fuse on or near microtubules.

**Figure 8—video 1.** Ribbon precursors attached to microtubules undergo fusion.
https://elifesciences.org/articles/98119/figures#fig8video1

**Figure 8—video 2.** Ribbon precursors attached to microtubules undergo fusion.
https://elifesciences.org/articles/98119/figures#fig8video2

**Figure 8—video 3.** Ribbon precursors attached to microtubules undergo fusion.
https://elifesciences.org/articles/98119/figures#fig8video3

could not be resolved separately and stayed together for the length of the remaining timelapse (or at least 5 min). Although we could not accurately measure the areas of precursors before and after fusion, we did observe that the relative area resulting from the fusion of two smaller precursors was greater than that of either precursor alone. This increase in the area suggests that precursor fusion may serve as a mechanism for generating larger ribbons (see examples: *Figure 8—figure supplement 1A–B*). Fusion events usually involved ribbon precursors on two separate microtubule filaments coming together during fusion, but fusion events could also occur on the same filament. Although the fusion events were infrequent, within our time windows (30–70 min), we quantified them and counted an average of 1.7 fusions and a maximum of five fusions events per neuromast in developing pLL hair cells (n=27 control neuromasts). Based on the close association with microtubules during these events, we tested whether an intact microtubule network was required for fusion events. We found that after treatment with 500 nM nocodazole for 30 min, there were fewer fusions events (*Figure 8C*, mean 0.4, maximum 2 fusions per neuromast, n=10 neuromasts). In contrast, the frequency of fusion events remained unchanged upon treatment with 25 µM taxol for 30 min (*Figure 8E*).

In our tracking analysis, we observed that Kif1aa was not required for the directional motion of ribbon and precursors in developing hair cells (*Figure 7E and H*). However, our immunohistological analyses indicated that there were more precursors and fewer complete synapses in *kif1aa* mutants. Therefore, we examined whether Kif1aa was important for fusion events. For this analysis, we quantified fusion events in *kif1aa* F0 crispants. We observed that, similar to nocodazole treatment, there were significantly fewer fusion events in *kif1aa* F0 crispants (*Figure 8E*, mean 0.6, maximum two fusions per neuromast, n=9 neuromasts). This reduction in fusion events may ultimately account for the excess of precursors and fewer complete synapses observed in *kif1aa* mutants (*Figure 5*). Together, our results indicate that during development, ribbon precursors can fuse when associated with microtubules, and an unperturbed microtubule network, along with Kif1aa is needed for these fusion events.

## Discussion

Our work in zebrafish applied high-resolution imaging approaches to investigate how ribbons are assembled and mobilized in developing hair cells. Our study demonstrates that hair cells have microtubule networks that are polarized with growing plus ends pointed to the cell base. Live imaging highlights that during development, ribbons and precursors show directed motion and fusion along microtubules and that an intact microtubule network is important for synapse formation.

### Comparison of ribbon synapse formation in zebrafish and mice

Compared to pLL hair cells (12–18 hr), synapse formation and refinement occur over a relatively long time window in mouse IHCs (E18-P14). In mice, much of the initial synaptogenesis occurs embryonically, while synapse refinement and pruning occur later, during the first postnatal week. Due to the rapid time course of synapse development in zebrafish, our experiments likely encompass ribbon precursor movement during synaptogenesis and synapse refinement. The companion paper on mouse IHCs focused primarily on ribbon mobility during the dramatic pruning of synapses that occurs during early postnatal development (*Voorn et al., 2024*). Unfortunately, it was not possible to study earlier events in mouse IHCs, as equivalent experiments were not possible embryonically. Live imaging at postnatal stages in mice indicates that during synapse pruning in IHCs, ribbons may detach from the membrane, undergo local trafficking, and fuse to nearby presynaptic AZs. Together, our studies in zebrafish and mouse hair cells illuminate several common features that characterize ribbon and microtubule dynamics during synapse formation.

First, immunostaining in both zebrafish and mice revealed an extensive network of microtubules within hair cells. In both species, the majority of microtubules were stabilized by tubulin acetylation (*Figure 2—figure supplement 1*). Our in vivo imaging of EB3-GFP in zebrafish hair cells revealed that microtubules are very dynamic, and the plus ends of microtubules grow towards the cell base, towards the presynaptic AZ (*Figure 2*). This data is consistent with data in mouse IHCs that demonstrated via immunostaining that the minus end marker of microtubules, CAMSAP2, localizes to the IHC apex (Vorn et al.). Overall, both studies demonstrate that microtubule networks in developing hair cells are polarized with the plus ends pointed to the cell base.

Second, live imaging in zebrafish and mouse hair cells revealed that ribbons associate with and move along microtubules. Tracking and quantification of ribbon movement via MSD analysis revealed that in both zebrafish and mouse hair cells, the majority of ribbons show motion indicative of confinement (*Figure 3*, $\alpha<1$). In addition, a subset of ribbon tracks shows evidence of directed motion (*Figure 3*, $\alpha>1$, or displacements >1 µm). In both species, after using nocodazole to destabilize microtubules, there was a dramatic reduction in instances of directed motion (*Figure 7*). In addition to ribbon movement, ribbon fusion was observed in the hair cells of mice and zebrafish (*Figure 8*). Interestingly, in mice, many fusion events were balanced out by divisions, or ribbons undergoing separation. In contrast, ribbon divisions were not prevalent in zebrafish hair cells. A lack of ribbon divisions could be due to the rapid speed of development in zebrafish hair cells, where divisions are too transient in nature to be confirmed. Alternatively, divisions may occur primarily late in synapse development and these events were overall less abundant in our zebrafish datasets. Importantly, in both mouse and zebrafish hair cells, ribbon fusion was diminished after microtubule destabilization. Together, these results in mice and zebrafish demonstrate that an intact microtubule network is required for the directed motion and fusion of ribbons during development.

Third, we sought to explore the kinesin motor responsible for ribbon movement in hair cells using zebrafish and mouse models. Based on scRNAseq data and previous immunostaining evidence, we focused on Kif1a (*Lush et al., 2019*; *Michanski et al., 2019*). In both mouse *Kif1a* and zebrafish *kif1aa* mutants, a common observation was an overall reduction in ribbon size (*Figure 5*). This suggests that a conserved mechanism where both intact microtubules and Kif1aa are required for normal ribbon formation. Despite these results, future work in zebrafish and mice is needed to explore additional kinesin motors that can drive ribbon movement during synapse formation.

## Fusion on microtubules as a mechanism for ribbon enlargement and maturation

Numerous studies have documented small ribbon precursors in developing photoreceptors and hair cells; later in development there are fewer, large ribbons present (*Michanski et al., 2019*; *Regus-Leidig et al., 2009*; *Schmitz, 2009*; *Sheets et al., 2011*; *Sobkowicz et al., 1986*; *Sobkowicz et al., 1982*). This has led to the hypothesis that ribbon precursor fusion is a mechanism to form larger, more mature ribbons (*Michanski et al., 2019*). Our live imaging experiments show that ribbon fusion is indeed a feature of ribbon formation (*Figure 8*, *Figure 8—figure supplement 1*). Interestingly, we observed that fusion often occurs between precursors on adjacent microtubules. Fusion events can occur both apically and basally within hair cells. More apical fusion events are likely a way to increase ribbon size early in development. More basal fusion events could represent the synapse reduction and refinement that occurs later in synapse development. Whether these later, basal fusion events occur after postsynaptic elements are present is unclear and awaits additional studies. Importantly, we found that the number of fusion events decreased when microtubule networks were disrupted or when Kif1aa was absent (*Figure 8*). Together, these results confirm the hypothesis that ribbons fuse during development. In addition, our work demonstrates that microtubules and Kif1aa are necessary for fusion and may help facilitate this process.

Ribbons are aggregates of proteins that share many features with biomolecular condensates that form through liquid-liquid phase separation (LLPS) (*Wang et al., 2021*). Biomolecular condensates are nano- to micro-meter scale compartments that function to concentrate proteins and nucleic acids. Some examples of biomolecular condensates include tight junctions, postsynaptic densities, ribonucleoprotein (RNP) granules, and stress granules (*Wang et al., 2021*; *Wiegand and Hyman, 2020*). These condensates are highly mobile and dynamic and constituent molecules diffuse readily and exchange with the surrounding environment. Biomolecular condensates often undergo fusion to form larger ones, and then shrink into a spherical shape (*Wiegand and Hyman, 2020*). FRAP analyses of Ribeye-GFP labeled ribbons in hair cells have confirmed that tagged Ribeye has a fast recovery phase within ribbons, verifying that Ribeye is highly mobile and dynamic within ribbons (*Graydon et al., 2017*). Furthermore, these FRAP studies revealed that Ribeye molecules can exchange with the surrounding environment. Our present studies confirmed that ribbon precursors undergo fusion to form larger spherical ribbons (*Figure 8A and B*). Together, these studies point towards the idea that precursors and ribbons are indeed biomolecular condensates. Ribeye also contains sequences predicted to be intrinsically disordered, a feature of molecules that can mediate condensate formation.

Interestingly, several biomolecular condensates such as RNP transport granules are known to be transported along the microtubules (*Knowles et al., 1996*), similar to our observations of ribbon precursor transport.

In our current study, we also found that microtubules are needed for fusion (*Figure 8C*). In other cellular contexts, the energy released by dynamic microtubules via growth and shrinkage can be used for force generation in a wide range of processes (*Vleugel et al., 2016*). Interestingly, work on stress granule condensates found a connection between dynamic microtubules and stress granule formation. This work demonstrated that microtubule growth and shrinkage promoted the fusion of small cytoplasmic granules into larger ones (*Chernov et al., 2009*). In addition, this study also demonstrated that stress granules slide along microtubules and that this movement may act in conjunction with pushing or pulling to promote fusion (*Chernov et al., 2009*). Ribbons and microtubules may also interact during development to promote fusion. Disrupting microtubules could interfere with this process, preventing ribbon maturation. Consistent with this, short-term (3–4 hr) and long-term (overnight) nocodazole increased ribbon and precursor numbers (*Figure 6G*; *Figure 4G*), suggesting reduced fusion. Long-term treatment (overnight) resulted in a shift toward smaller ribbons (*Figure 4H–I*), and ultimately fewer complete synapses (*Figure 4F*).

We also observed that fusion was disrupted in *kif1aa* mutants, despite finding that directed transport of precursors and ribbons remained unchanged in *kif1aa* mutants (*Figures 7 and 8*). A recent study on RNP condensates examined kinesin motors and adaptors in the context of microtubules (*Cochard et al., 2023*). This work found that the precise combination of motor proteins or adaptors can impact where condensates form. Some combinations enabled the formation and transport of condensates along microtubules. However, other combinations restricted condensate formation to microtubule terminals where motor proteins are predicted to form a scaffold. Our work indicates that while more than one kinesin motor protein is capable of transporting ribbon precursors, it is predominantly Kif1aa that mediates fusion along microtubules. This could explain why precursor fusion, but not transport is impacted in *kif1aa* mutants (*Figures 7 and 8*). This scenario could also account for the increased number of precursors and the reduced number of complete synapses observed in *kif1aa* mutants (*Figure 5*).

## Kinesin motors and adaptors for ribbon transport on microtubules

Our immunohistochemistry analyses show clear synaptic defects in *kif1aa* mutants after the bulk of synapse formation has occurred (fewer synapses, more precursors, *Figure 5*). Despite these defects, we were unable to demonstrate that loss of Kif1aa impacts the movement of precursors along microtubules (*Figure 7*). These results suggest that another kinesin motor may function to transport precursors and ribbons. Pulldown assays have shown an interaction of Kif3a with Ribeye (*Uthaiah and Hudspeth, 2010*). While *kif1aa* mRNA is the most abundant kinesin motor protein transcript detected in pLL hair cells, *kif3a* mRNA is also present. *Kif3a* mRNA is present at lower levels and is expressed more broadly in pLL hair cells, supporting cells, and afferent neurons (*Sur et al., 2023*). Therefore, both Kif1aa and Kif3a may be competent to transport developing ribbons. Future work exploring the role of Kif3a separately or in combination with Kif1aa will illuminate the role these motors play in synapse assembly in hair cells. In addition, it will be useful to visualize these kinesins by fluorescently tagging them in live hair cells to observe whether they associate with ribbons.

Regardless of the anterograde kinesin motor(s) that transports precursors to the cell base, we also observed that precursors can also move in the retrograde direction, towards the cell apex (*Figure 3C* bottom panels, *Figure 3*, *Figure 3—figure supplement 1*). In fact, out of the tracks with directional motion, we found that 20.8% of precursors moved apically. This could be due to the fact that a subpopulation of microtubules shows plus-end mediated growth apically (20.4 %). Alternatively, it is possible that retrograde motors may also transport ribbon precursors. For example, this could be accomplished using minus-end directed motors, such as cytoplasmic dynein heavy chains, or kinesins in the kinesin-14 family (*Sweeney and Holzbaur, 2018*; *Yamada et al., 2017*). In the future, it will be important to explore the role of these additional motor proteins in the context of ribbon and precursor mobility.

Our findings indicate that ribbons and precursors show directed motion indicative of motor-mediated transport (*Figures 3 and 7*). While a subset of ribbons moves directionally with α values >1, canonical motor-driven transport in other systems, such as axonal transport, can achieve even

higher α values approaching 2 (*Bellotti et al., 2021*; *Corradi et al., 2020*). We suggest that relatively lower α values arise from the highly dynamic nature of microtubules in hair cells. In axons, microtubules form stable, linear tracks that allow kinesins to transport cargo with high velocity. In contrast, the microtubule network in hair cells is highly dynamic, particularly near the cell base. Within a single time frame (50–100 s), we observe continuous movement and branching of these networks. This dynamic behavior adds complexity to ribbon motion, leading to frequent stalling, filament switching, and reversals in direction. As a result, ribbon transport appears less directional than the movement of traditional motor cargoes along stable axonal filaments, resulting in lower α values compared to canonical motor-mediated transport. Notably, treatment with taxol, which stabilizes microtubules, increased α values to levels closer to those observed in canonical motor-driven transport (*Figure 7—figure supplement 1*). This finding supports the idea that the relatively lower α values in hair cells are a consequence of a more dynamic microtubule network. Overall, this dynamic network gives rise to a slower, non-canonical mode of transport.

Another important component of motor-mediated transport is adaptor proteins that selectively link cargo to specific motor proteins (*Fu and Holzbaur, 2014*). In neurons, cargos leave the Golgi apparatus and are packed into specialized transport vesicles, along with adaptor scaffolds (*Farías et al., 2012*; *Sampo et al., 2003*). Work in neurons has shown that the Kif1a motor transports Rab3-positive synaptic vesicle precursors (*Okada et al., 1995*). This transport is thought to rely on DENN (differentially expressed in normal and neoplastic cells)/MADD (MAP kinase activating death domain) which links synaptic vesicle precursors to Kif1a (*Hummel and Hoogenraad, 2021*; *Niwa et al., 2008*). Interestingly, like mature ribbons, ribbon precursors are decorated with synaptic vesicles (*Michanski et al., 2019*). Thus, is it possible that these synaptic vesicles may contain adaptor proteins that couple the ribbon to a kinesin motor to enable transport. In the future, it will be important to identify this adaptor protein and to understand what other molecules are co-transported with Ribeye during ribbon formation.

## Ribbon formation in the absence of microtubules

Our work indicates that over short time scales (30–70 min), microtubule destabilization via nocodazole treatment reduces the number of ribbons and precursors that show directed motion indicative of active transport (*Figure 7*). In addition, prolonged nocodazole treatment throughout development (16 hr) leads to the formation of fewer synapses (*Figure 4*). But in both treatment paradigms, some ribbons still show directed motion, and some synapses continue to form despite microtubule disruption. If tracks of microtubules are required for developing precursor and ribbon mobility during development, what underlies this residual movement and synapse formation?

One possibility is that not all microtubules are disrupted after nocodazole treatment. While high doses of nocodazole (~40 μM) eliminate all microtubules, they are also cytotoxic (*Laisne et al., 2021*). The doses used in our experiments (100–500 nM) were not cytotoxic, although higher doses did result in the death of hair cells. At these lower doses, microtubules—especially the more apical, stable population (see examples: *Figure 2—figure supplement 1B*, *Figure 4—figure supplement 1F–G*)—were not entirely disrupted. Residual ribbon mobility during nocodazole treatment could occur along these remaining microtubules. Alternatively, other cytoskeletal components, such as actin or intermediate filaments may contribute to ribbon mobility. Work in mouse IHCs has shown that actin filaments help regulate and organize synaptic vesicles at mature ribbon synapses (*Guillet et al., 2016*). In addition, in mice, the actin-based motor protein Myosin6 has been implicated in ribbon-synapse formation and function (*Roux et al., 2009*). A third possibility is that ribbons and precursors move through the cytoplasm via diffusion and are captured at the AZ via an adapter protein. Although we primarily observed ribbons and precursors attached to microtubules, when these filaments are destabilized using nocodazole, movement could still occur via diffusion. Once near the base of the cell, presynaptic molecules such as Bassoon could act to anchor ribbons at the AZ (*Jing et al., 2013*). Any of these scenarios—residual microtubule-based transport, alternative cytoskeletal component, or diffusion followed by capture—could explain how ribbons continue to move and how synapses still form in the absence of intact microtubules. In all likelihood, a combination of these processes is required to ensure that ribbon synapses form properly. Additional pharmacological and imaging experiments are needed to delineate the relative contributions of these processes.

Another important consideration is the potential off-target effects of nocodazole. Even at non-cytotoxic doses, nocodazole toxicity may impact ribbons and synapses independently of its effects on microtubules. While this is less of a concern in the short- and medium-term experiments (30 min to 4 hr), long-term treatments (16 hr) could introduce confounding effects. Additionally, nocodazole treatment is not hair cell-specific and could disrupt microtubule organization within afferent terminals as well. Thus, the reduction in ribbon-synapse formation following prolonged nocodazole treatment may result from microtubule disruption in hair cells, afferent terminals, or a combination of the two.

### Does spontaneous activity shape ribbon transport or fusion?

In previous work, we demonstrated that spontaneous calcium activity impacts ribbon formation in pLL hair cells (*Sheets et al., 2012*; *Wong et al., 2019*). Spontaneous activity is well documented in developing sensory systems [reviewed in: *Leighton and Lohmann, 2016*]. In the inner ear of mammals, spontaneous activity in sensory hair cells is thought to act during development to establish neuronal connections within the inner ear, and downstream in the brain to shape tonotopic maps (*Ceriani et al., 2019*; *Tritsch et al., 2007*). In our previous work in pLL hair cells, we found that spontaneous rises in presynaptic calcium loads calcium into synaptic mitochondria. Calcium loading into the mitochondria regulates the amount of $NAD^+$ to NAD(H) in the developing hair cell. Blocking either presynaptic or mitochondria calcium during development results in higher levels of $NAD^+$, the formation of larger ribbons, fewer synapses, and the retention of small ribbon precursors in hair cells (*Sheets et al., 2012*; *Wong et al., 2019*). What aspect of ribbon formation impacted by spontaneous activity remains unclear.

Work in yeast has shown that calcium is important for microtubule stability (*Adamíková et al., 2004*; *Façanha et al., 2002*). In addition, work in dendritic spines has shown that synaptic calcium responses promote microtubule entry into active spines (*Merriam et al., 2013*). Therefore, it is possible that spontaneous calcium activity may act to stabilize microtubules or facilitate movement to the presynaptic AZ to facilitate ribbon transport. In addition to altering the microtubule network, spontaneous activity could also impact ribbon fusion. Protein condensate formation and fusion are influenced by many aspects of the cellular environment, including temperature, pH, osmolarity, and ion concentration (*Wang et al., 2022*). Elevated calcium can promote the fusion of chromogranin proteins that undergo LLPS in the Golgi lumen (*Gerdes et al., 1989*; *Yoo, 1995*). Therefore, it is possible that elevated calcium during a spontaneous calcium event could promote ribbon or precursor fusion. In addition to the cellular environment, post-translational modifications to the proteins within the condensate can impact formation and LLPS. Such modifications include: phosphorylation, acetylation, SUMOylation, ubiquitination, methylation, and ADP-ribosylation (*Luo et al., 2021*). These modifications can alter protein-protein interactions by changing the charge, structure, or hydrophobicity. Our previous work indicated that $NAD^+$ or NAD(H) via a $NAD^+$/NADH binding domain on Ribeye impacts ribbon formation (*Wong et al., 2019*). It is possible that in addition to calcium, $NAD^+$/NADH levels in the cell could modify Ribeye to facilitate condensate formation or fusion. In the future, it will be important to use the live imaging approaches outlined here to understand how spontaneous presynaptic and mitochondrial calcium influx impact the movement and fusion of ribbon precursors.

Overall, our live imaging studies demonstrate that microtubule networks are critical for ribbon and precursor mobility, and fusion in developing zebrafish hair cells. However, ribbon synapses contain many molecules, and synapse formation requires many successive steps. In the future, it will be important to develop approaches to endogenously tag and label other presynaptic ($Ca_V1.3$ channels, Bassoon, Piccolino) and postsynaptic proteins (PSD95, GluR2/3/4) that make up ribbon synapses. For this future work, zebrafish is an ideal model system for creating these new genetic tools and for live imaging. By imaging ribbons, along with these other tagged synaptic components, we will gain a more comprehensive picture of synapse formation. Understanding how ribbon synapses form is essential to determining how to reform synapses when they are disrupted in auditory and visual disorders.

# Methods

### Key resources table

| Reagent type (species) or resource | Designation | Source or reference | Identifiers | Additional information |
|---|---|---|---|---|
| Strain, strain background (*D. rerio*) | Tübingen zebrafish; TU | Other | Background stain | ZFIN:ZDB-GENO-990623–3 |
| Genetic reagent (*D. rerio*) | *Tg(myo6b:ctbp2a-TagRFP)*idc11Tg | *Wong et al., 2019* | Transgenic line | ZFIN: ZDB-ALT-190102–4 |
| Genetic reagent (*D. rerio*) | *Tg(myo6b:YFP-Hsa.TUBA)*idc16Tg | *Ohta et al., 2020* | Transgenic line | ZFIN: ZDB-ALT-210824–2 |
| Genetic reagent (*D. rerio*) | *myo6b:EB3-GFP*idc23Tg | This paper | Transgenic line | ZFIN: ZDB-ALT-230928–1 |
| Genetic reagent (*D. rerio*) | *kif1aa*idc24 | This paper and (*David et al., 2024*) | Mutant strain | ZFIN: ZDB-ALT-240416–3 |
| Transfected construct (plasmid, injected) | *myo6b:EB3-GFP* | This paper | Tol2 transgenesis construct | ZFIN: ZDB-TGCONSTRCT-230928–1 |
| Recombinant DNA reagent | p5E-*pmyo6b* | *Trapani et al., 2009* | Gateway entry clone | |
| Recombinant DNA reagent | pME-*EB3-GFP* | *Kawano et al., 2022* | Gateway entry clone | |
| Recombinant DNA reagent | pDestTol2pACryGFP | *Berger and Currie, 2013* | Addgene plasmid # 64022, RRID:Addgene_64022 | |
| Recombinant DNA reagent | p3E-*polyA* | *Kwan et al., 2007* | Gateway entry clone | |
| Sequence-based reagent | *kif1aa* gRNA | This paper | gRNA | 5'-ACGGATGTTCTCGCACACGT(AGG)–3' |
| Sequence-based reagent | *kif1aa* gRNA | This paper | gRNA | 5'-GTGCGAGAACATCCGTTGCT(AGG)–3' |
| Sequence-based reagent | *kif1aa* gRNA | This paper | gRNA | 5'-TGGACTCCGGGAATAAGGCT(AGG)–3' |
| Sequence-based reagent | *kif1aa* gRNA | This paper | gRNA | 5'-AGAATACCTAGCCTTATTCC(CGG)–3'. |
| Sequence-based reagent | *kif1aa*_FWD | This paper | PCR primers | 5'-AACACCAAGCTGACCAGTGC-3' |
| Sequence-based reagent | *kif1aa*_REV | This paper | PCR primers | 5'-TGCGGTCCTAGGCTTACAAT-3' |
| Sequence-based reagent | *kif1aa*_FWD_fPCR | This paper | PCR primers | 5'-TGTAAAACGACGGCCAGT-AAATAGAGATTCACTTTTAATC-3' |
| Sequence-based reagent | *kif1aa*_REV_fPCR | This paper | PCR primers | 5'- GTGTCTT-CCTAGGCTTACAATGCTTTTGG-3' |
| Antibody | anti-Myosin-VIIa (rabbit polyclonal) | Proteus Biosciences | Cat# 25–6790, RRID:AB_10015251 | IF(1:1000) |
| Antibody | anti-pan-Maguk (mouse monoclonal IgG1) | Millipore | Cat# MABN72, RRID:AB_10807829 | IF(1:500) |
| Antibody | anti-GFP (chicken polyclonal) | Aves labs | Cat# GFP-1010, RRID:AB_10000240 | IF(1:1000) |
| Antibody | anti-tyrosinated-α-tubulin (mouse monoclonal IgG2a) | Sigma Aldrich | Cat# MAB1864-I, RRID:AB_2890657 | IF(1:1000) |
| Antibody | anti-acetylated-α-tubulin (mouse monoclonal IgG2b) | Sigma Aldrich | Cat# T7451, RRID:AB_609894 | IF(1:5000) |
| Antibody | anti-Ribeyeb (mouse monoclonal IgG2a) | *Sheets et al., 2011* | | IF(1:10,000) |
| Antibody | anti-CTPB (mouse monoclonal IgG2a) | Santa Cruz | Cat# sc-55502, RRID:AB_629339 | IF(1:1000) |

*Continued on next page*

*Continued*

| Reagent type (species) or resource | Designation | Source or reference | Identifiers | Additional information |
|---|---|---|---|---|
| Antibody | Anti-mouse secondary antibodies (goat polyclonal) | Thermo Fisher Scientific | Cat# A-2114; RRID:AB_2535779 | IF(1:1000) |
| Antibody | Anti-mouse secondary antibodies (goat polyclonal) | Thermo Fisher Scientific | Cat# A-21143 RRID:AB_2535779 | IF(1:1000) |
| Antibody | Anti-mouse secondary antibodies (goat polyclonal) | Thermo Fisher Scientific | Cat# A-2113; RRID:AB_2535771 | IF(1:1000) |
| Antibody | Anti-mouse secondary antibodies (goat polyclonal) | Thermo Fisher Scientific | Cat# A-21240; RRID:AB_2535809 | IF(1:1000) |
| Antibody | Anti-mouse secondary antibodies (goat polyclonal) | Thermo Fisher Scientific | Cat# A-21242; RRID:AB_253581 | IF(1:1000) |
| Antibody | Anti-mouse secondary antibodies (goat polyclonal) | Thermo Fisher Scientific | Cat# A-2124; RRID:AB_2535810 | IF(1:1000) |
| Antibody | Anti-rabbit secondary (goat polyclonal) | Thermo Fisher Scientific | Cat# A-11008; RRID:AB_143165, | IF(1:1000) |
| Antibody | Anti-chicken secondary (goat polyclonal) | Thermo Fisher Scientific | Cat# A-11039, RRID:AB_2534096 | IF(1:1000) |
| Commercial assay, kit | LIZ500, fPCR dye standard | Applied Biosystems | Cat# 4322682 | |
| Peptide, recombinant protein | Bs1I | New England Biolabs | Cat# R0555S | |
| Peptide, recombinant protein | Cas9 protein | Integrated DNA technologies | Cat# 1081059 | |
| Chemical compound, drug | ethyl 3-aminobenzoate methanesulfonate salt | Sigma Aldrich | Cat# A5040 | |
| Chemical compound, drug | nocodazole | Sigma Aldrich | Cat# SML1665 | 250–500 nM |
| Chemical compound, drug | Paclitaxel | Sigma Aldrich | Cat# 5082270001 | 25 µM |
| Software, algorithm | Imaris 9.9 | Oxford Instruments | RRID:SCR_007370 | |
| Software, algorithm | Matlab R2020b | Mathworks | RRID:SCR_001622 | |
| Software, algorithm | MSDanalyzer | *Tarantino et al., 2014* | | |
| Software, algorithm | Prism 10 | Graphpad | RRID:SCR_002798 | |
| Software, algorithm | FIJI | Open source | RRID:SCR_002285 | |

## Zebrafish animals

Zebrafish (*Danio rerio*) were bred and cared for at the National Institutes of Health (NIH). This research was approved by the NINDS/NIDCD/NCCIH animal care and use committee (ACUC) under animal study protocol #1362–13. Zebrafish larvae were raised at 28 °C in E3 embryo medium (5 mM NaCl, 0.17 mM KCl, 0.33 mM CaCl$_2$, and 0.33 mM MgSO$_4$, buffered in HEPES, pH 7.2). All experiments were performed on larvae aged 2–5 days post fertilization (dpf). Larvae were chosen at random at an age where sex determination is not possible. The previously described mutant and transgenic lines were used in this study: *Tg(myo6b:ctbp2a-TagRFP)*[idc11Tg] referred to as *myo6b:riba-TagRFP*; *Tg(myo-6b:YFP-Hsa.TUBA)*[idc16Tg] referred to as *myo6b:YFP-tubulin* (*Ohta et al., 2020*; *Wong et al., 2019*). *Tg(myo6b:ctbp2a-TagRFP)*[idc11Tg] reliably labels mature ribbons, similar to a pan-CTBP immunolabel at 5 dpf (*Figure 1—figure supplement 1A–B*). This transgenic line does not alter the number of hair cells or complete synapses per hair cell (*Figure 1—figure supplement 1A–D*). In addition, *myo6b:ctbp2a-TagRFP* does not alter the size of ribbons (*Figure 1—figure supplement 1E*).

## Zebrafish transgenic and CRISPR-Cas9 mutant generation

To create *myo6b:EB3-GFP* transgenic fish, plasmid construction was based on the tol2/gateway zebrafish kit (*Kwan et al., 2007*). The p5E *pmyo6b* entry clone was used to drive expression in hair

cells. A pME-*EB3-GFP* clone was kindly provided by Catherine Drerup at the University of Wisconsin, Madison. pDestTol2pACryGFP was a gift from Joachim Berger & Peter Currie (Addgene plasmid # 64022). These clones were used along with the following tol2 kit gateway clone, p3E-*polyA* (#302) to create the expression construct: *myo6b:EB3-GFP*. To generate the stable transgenic fish line *myo6b:EB3-GFP*[idc23Tg], plasmid DNA, and tol2 transposase mRNA were injected into zebrafish embryos as previously described (*Kwan et al., 2007*). The *myo6b:EB3-GFP*[idc23Tg] transgenic line was selected for a single copy and low expression of EB3-GFP.

A *kif1aa* germline mutant (*kif1aa*[idc24]) was generated using CRISPR-Cas9 technology as previously described (*Varshney et al., 2016*). Exon 6, containing part of the Kinesin motor domain was targeted (*Figure 5—figure supplement 1A*). Guides RNAs (gRNAs) targeted to *kif1aa* are as follows: 5′-ACGG ATGTTCTCGCACACGT(AGG)–3′, **5′-**GTGCGAGAACATCCGTTGCT(AGG)–3′, 5′-TGGACTCCGGGA ATAAGGCT(AGG)–3′, 5′-AGAATACCTAGCCTTATTCC(CGG)–3′. Founder fish were identified using fragment analysis of fluorescent PCR (fPCR) products. A founder fish containing a complex insertion or deletion (INDEL) that destroys a BslI restriction site in exon 6 was selected (*Figure 5—figure supplement 1B*). This INDEL disrupts the protein at amino acid 166 (*Figure 5—figure supplement 1A*). Subsequent genotyping was accomplished using standard PCR with touchdown, and BslI restriction enzyme digestion. *Kif1aa* genotyping primers used were: *kif1aa*_FWD 5′-AACACCAAGCTGACCA GTGC-3′ and *kif1aa*_REV 5′-TGCGGTCCTAGGCTTACAAT-3′.

Because *kif1aa* mutants have no phenotype to distinguish them from sibling controls at the ages imaged, the low throughput of our live imaging approaches made using germline mutants prohibitive. Therefore, we created *kif1aa* F0 crispants for our live imaging analyses. Here, we injected the following *kif1aa* gRNAs: 5′-GTGCGAGAACATCCGTTGCT(AGG)–3′ and 5′-AGAATACCTAGCCTTA TTCC(CGG)–3′, along with Cas9 protein, as previously described (*Hoshijima et al., 2019*). We then grew *kif1aa*-injected F0 crispants for 2 d and then used them for our live imaging analyses. Studies have shown that F0 crispants are a fast and effective way to knock down gene function in any genetic background (*Hoshijima et al., 2019*; *Sheets et al., 2021*). After live imaging, we genotyped all *kif1aa* F0 crispants (*Figure 6—figure supplement 1*) to ensure that the gRNAs cut the target robustly using fPCR and the following primers: *kif1aa*_FWD_fPCR 5′-TGTAAAACGACGGCCAGT-AAATAGAGATTC ACTTTTAATC-3′ and *kif1aa*_REV_fPCR 5′- GTGTCTT-CCTAGGCTTACAATGCTTTTGG-3′ (*Carrington et al., 2015*). fPCR fragments were run on a genetic analyzer (Applied Biosystems, 3500XL) using LIZ500 (Applied Biosystems, 4322682) as a dye standard. Analysis of fPCR revealed an average peak height of 4740 a.u. in wild type, and an average peak height of 126 a.u. in *kif1aa* F0 crispants (*Figure 6—figure supplement 1E–F*). Any *kif1aa* F0 crispant without robust genomic cutting or a peak height >500 a.u. was not included in our analyses.

## Zebrafish pharmacology

To destabilize or stabilize microtubules, larval zebrafish at 2 dpf were incubated in either nocodazole (Sigma-Aldrich, SML1665) or Paclitaxel (taxol) (Sigma-Aldrich, 5082270001). Both drugs were maintained in DMSO. For experiments, these drugs were diluted in media for a final concentration of 0.1% DMSO, 250–500 nM nocodazole, and 25 µM taxol. For controls, larvae were incubated in media containing 0.1% DMSO. For long-term incubation (16 hr), wild-type larvae were incubated in E3 media containing 250 nM nocodazole or 25 µM taxol at 54 hpf for 16 hr (overnight). After this long-term treatment, larvae were fixed and prepared for immunohistochemistry (see below). For live, short-term incubations (for 3–4 hr incubations or ribbon tracking), transgenic larvae (*myo6b:riba-TagRFP*; *myo6b:YFP-tubulin*) at 48–54 hpf were embedded in 1% low melt agarose prepared in E3 media containing 0.03% tricaine (Sigma-Aldrich, A5040, ethyl 3-aminobenzoate methanesulfonate salt). 500 nM nocodazole, 25 µM taxol, or DMSO were added to the agarose and to the E3 media used to hydrate the sample. For short-term treatments, hair cells were imaged after 30 min of embedding.

## Immunohistochemistry of zebrafish samples

Immunohistochemistry used to label acetylated-α-tubulin, tyrosinated-α-tubulin, Ribeyeb, or pan-CTBP (ribbons and precursors), pan-Maguk (postsynaptic densities), and Myosin7a (cell bodies) was performed on whole zebrafish larvae similar to previous work. The following primary antibodies were used: rabbit anti-Myosin7a (Proteus 25–6790; 1:1000); mouse anti-pan-Maguk (IgG1) (Millipore MABN72; 1:500); mouse anti-Ribeyeb (IgG2a) (*Sheets et al., 2011*; 1:10,000); mouse anti-CTPB

(IgG2a) (Santa Cruz sc-55502; 1:1000); mouse anti-acetylated-α-tubulin (IgG2b) (Sigma-Aldrich T7451; 1:5000); mouse anti-tyrosinated-α-tubulin (IgG2a) (Sigma-Aldrich MAB1864-I; 1:1000); chicken anti-GFP (to stain YFP-tubulin) (Aves labs GFP-1010; 1:1,000). The following secondary antibodies were used at 1:1000: (Thermo Fisher Scientific, A-11008; A-21143, A-21131, A-21240, A-21242, A-21241, A-11039). Larvae were fixed with 4% paraformaldehyde in PBS for 4 hr at 4 °C. All wash, block, and antibody solutions were prepared in 0.1% Tween in PBS (PBST). After fixation, larvae were washed 5×5 min in PBST. Prior to block, larvae were permeabilized with acetone. For this permeabilization, larvae were first washed for 5 min with $H_2O$. The $H_2O$ was removed and replaced with ice-cold acetone and samples were placed at −20 °C for 3 min, followed by a 5 min $H_2O$ wash. The larvae were then washed for 5×5 min in PBST. Larvae were then blocked overnight at 4 °C in blocking solution (2% goat serum, 1% bovine serum albumin, and 2% fish skin gelatin in PBST). Larvae were then incubated in primary antibodies in antibody solution (1% bovine serum albumin in PBST) overnight, nutating at 4 °C. The next day, the larvae were washed for 5×5 min in PBST to remove the primary antibodies. Secondary antibodies in antibody solution were added and larvae were incubated for 3 hr at room temperature. After 5×5 min washes min in PBST to remove the secondary antibodies, larvae were rinsed in $H_2O$ and mounted in Prolong Gold (Thermo Fisher Scientific, P36930).

## Confocal imaging and analysis of fixed zebrafish samples

After immunostaining, fixed zebrafish samples were imaged on an inverted Zeiss LSM 780 (Zen 2.3 SP1) or an upright Zeiss LSM 980 (Zen 3.4) laser-scanning confocal microscope with Airyscan using a 63x1.4 NA oil objective lens. Z-stacks encompassing the entire neuromast were acquired every 0.17 (LSM 980) or 0.18 (LSM 780) μm with a 0.04 μm x-y pixel size and Airyscan autoprocessed in 3D.

Synaptic images from fixed samples were further processed using FIJI. Acetylated-α-tubulin or Myosin7 label was used to manually count hair cells. Complete synapses comprised of both a Ribeyeb/CTBP and Maguk puncta were also counted manually. To quantify the area of each ribbon and precursor, images were processed in FIJI using a macro, 'IJMacro_AIRYSCAN_simple3dSeg_ribbons only.ijm' as previously described (*Hussain et al., 2025*; *Wong et al., 2019*). Here, each Airyscan z-stack was max-projected, and background corrected using rolling-ball subtraction. A threshold was applied to each image, followed by segmentation to delineate individual Ribeyeb/CTBP puncta. The watershed function was used to separate adjacent puncta. A list of 2D objects of individual ROIs (minimum size filter of 0.002 μm$^2$) was created to measure the 2D areas of each Ribeyeb/CTBP puncta. Areas for all Ribeyeb/CTBP puncta within each neuromast were then exported as a csv spreadsheet. Areas were averaged per neuromast, per genotype, or plotted in a frequency distribution. For comparisons, all fixed images analyzed in FIJI were imaged and processed using the same parameters.

To quantify the mean intensity of acetylated-α-tubulin after overnight nocodazole or taxol treatments, 20 slices centered on the hair cells were max-projected in FIJI. An ROI was drawn around the hair cells, and this ROI was used to measure the mean intensity of the acetylated-α-tubulin label in each neuromast.

## Confocal imaging and in vivo analysis of ribbon numbers in developing zebrafish hair cells

For counting ribbon numbers in developing and mature hair cells (*Figure 1*), double transgenic *myo6b:riba-TagRFP* and *myo6b:YFP-tubulin* larvae at 2 and 3 dpf were imaged. Transgenic larvae were pinned to a Sylgard-filled petri dish in E3 media containing 0.03% tricaine and imaged on a Nikon A1R upright confocal microscope using a 60x1 NA water objective lens. Denoised images were acquired using NIS Elements AR 5.20.02 with a 0.425 μm z-interval, at 16 x averaging, and 0.05 μm/pixel. Z-stacks of whole neuromasts, including the kinocilium were acquired in a top-down configuration using 488 and 561 nm lasers. The 488 nm laser along with a transmitted PMT (T-PMT) detector was used to capture the kinocilial heights.

For the quantification of ribbon numbers at different developmental stages (*Figure 1*), a custom-written Fiji macro 'Live ribbon counter' was used to batch-process the z-stacks (*Hussain et al., 2025*). The red channel (Riba-TagRFP) of each z-stack was thresholded (threshold value = 97). Watershed was applied to the thresholded stack to separate ribbons near each other. The resulting mask from the thresholding and water shedding was applied to the original red channel. The number of ribbons was then counted using '3D Objects Counter' plugin (Threshold = 1, min size = 0, max size = 183,500).

The counted objects were merged with the green channel (YFP-tubulin). Each z-stack was visually inspected to determine the localization of the ribbons. Ribbons below the nucleus were classified as 'basal' and the rest as 'apical.' The number of apical and basal ribbons was counted in each hair cell.

To classify the developmental stage of each hair cell (*Figure 1*), the height of the kinocilium was used. The number of z-slices between the kinocilium tip and base was determined and multiplied by the z-slice interval (0.425 µm) to get the kinocilium height. Hair cells with heights <1.5 µm were classified as 'early', heights 1.5–10 µm were classified as 'intermediate', and heights 10–18 µm were classified as 'late.' Hair cells with heights >18 µm were considered 'mature'.

## Confocal imaging and in vivo tracking EB3-GFP dynamics in zebrafish

Transgenic *myo6b:EB3-GFP* larvae at 2–3 dpf were mounted in 1% LMP agarose containing 0.03% tricaine in a glass-bottom dish. Larvae were imaged on an inverted Zeiss LSM 780 (Zen 2.3 SP1) confocal microscope using a 63×1.4 NA oil objective lens. For timelapses, confocal z-stacks of partial cell volumes (3.5 µm, 7 z slices at 0.5 µm z interval) with a 0.07 µm x-y pixel size were taken every 7 s for 15–30 min.

The EB3-GFP timelapses were registered in FIJI using the plugin 'Correct 3D drift' (*Parslow et al., 2014*), max-projected, and then tracked in 2D in Imaris. For spot detection, we used an estimated xy diameter of 0.534 µm with background subtraction. The detected spots were filtered by 'Quality' using the automatic threshold. The timelapses were visually checked to make sure the spot detection was accurate. For the tracking step, the 'Autoregressive motion' algorithm was used, with a maximum linking distance of 1 µm and a maximum gap size of three frames. To ensure accurate track detection, short tracks were removed by filtering for the number of spots in a track (>5) and track displacement length (>automatic threshold).

To calculate the track angles relative to the cell base, we used cells that lie horizontally, so we only needed to consider the angles in the 2D, xy plane. In Imaris, the tracks in each hair cell were selected and exported separately. Using the start and end position coordinates of the exported tracks, we calculated track angles in MATLAB using custom-written code called, 'EB3 track angle' (*Hussain et al., 2025*). The angle of each hair cell was measured in Imaris. The final track angle distribution plotted was obtained by measuring the difference between each track angle and the angle of the hair cell.

To create movies of EB3-GFP tracks in *Figure 2—video 1*, the FIJI plugin 'Correct 3D drift' was applied to the timelapse. Z-stacks were then max-projected, and tracks were detected using the FIJI plugin TrackMate (*Parslow et al., 2014*; *Tinevez et al., 2017*). For *Figure 2—video 1*, the LoG detector in TrackMate was used with an estimated object diameter of 0.6 µm, and a quality threshold of 8, using a median filter and sub-pixel localization. The Linear Assignment Problem (LAP) tracker was selected using a frame-to-frame linking max distance of 1 µm, a track segment gap closing max distance of 1 µm and a max frame gap of 2 µm. Tracks were colored by Track index. For viewing tracks over time, tracks were displayed as 'Show tracks backwards in time' with a fade range of 5-time points. To create a color-coded temporal map of EB3-GFP tracks over a short time window (21 s, *Figure 2C–D*), the FIJI Hyperstack plugin 'Temporal-Color code' was used with the 16 colors LUT.

## Confocal imaging and in vivo analysis of ribbon numbers after short-term pharmacological treatments in zebrafish

For counting ribbons after 3–4 hr drug treatment, transgenic zebrafish expressing *myo6b:riba-TagRFP* and *myo6b:YFP-tubulin* at 2 dpf were examined. Transgenic larvae were mounted in 1% low melt agarose in E3 media containing 0.03% tricaine and one of the following: 500 nM nocodazole, 25 µM taxol, or 0.1% DMSO (control). Samples were imaged on an inverted Zeiss LSM 780 (Zen 2.3 SP1) confocal microscope with Airyscan, along with a 63 x NA 1.4 oil objective lens. Z-stacks encompassing the entire neuromast were acquired every 0.18 µm with a 0.04 µm x-y pixel size and Airyscan auto-processed in 3D.

To quantification of ribbon numbers before and after 3–4 hr nocodazole and taxol treatment or in *kif1aa* F0 crispants (*Figure 6*, *Figure 6—figure supplement 2*), the custom-written Fiji macro 'Live ribbon counter' described above was used to batch-processed the z-stacks (*Hussain et al., 2025*). The red channel (Riba-TagRFP) of each z-stack was thresholded (threshold value = 28) and segmented (watershed). The resulting mask was applied to the original red channel. The number of ribbons was then counted using '3D Objects Counter' (threshold = 1, min size = 0, max size = 183,500).

The counted objects were merged with the green channel (YFP-tubulin). Each z-stack was visually inspected to make sure the objects counted were within hair cells. The number of ribbons per neuromast was determined and divided by the number of hair cells. The difference in ribbon numbers pre- and post-drug treatment was plotted.

## Confocal imaging and in vivo tracking of ribbons

To visualize ribbon precursor movement, timelapses of double transgenic *myo6b:riba-TagRFP* and *myo6b:YFP-tubulin* larvae at 2 dpf were imaged. For pharmacological treatments, transgenic larvae were mounted in 1% low melt agarose in E3 media containing 0.03% tricaine and one of the following: 500 nM nocodazole, 25 μM taxol, or 0.1% DMSO (control) in a glass-bottom dish. Double transgenic *kif1aa* F0 crispants and uninjected controls were mounted in 1% low melt agarose in E3 media containing 0.03% tricaine in a glass-bottom dish. Larvae were imaged on an inverted Zeiss LSM 780 or an upright LSM 980 confocal microscope with Airyscan using a 63×1.4 NA oil objective lens. Airyscan z-stacks of partial cell volumes (~3 μm, 15–20 z-slices using 0.18 μm z-interval and a 0.04 μm x-y pixel size) were taken on the LSM 780 every 50–100 s for 30–70 min. Faster LSM 980 Airyscan z-stacks of partial cell volumes (~2–3.5 μm, 12–20 z-slices using 0.17 μm z interval and a 0.04 μm x-y pixel size) were taken every 3–20 s for 5–40 min. Airyscan timelapses were autoprocessed in 3D. In addition, we acquired a subset of LSM 780 Airyscan z-stacks every 5–8 min for 30–100 min to capture fusion events more clearly for *Figure 8—videos 1, 2 and 3*.

Timelapses were registered using the FIJI plugin 'Correct 3D drift.' Drift-corrected timelapses were then tracked in 3D in Imaris using spot detection with estimated xy diameters of 0.427 μm (with background subtraction). The spots were filtered based on 'Quality,' with thresholds between 3–8, chosen after visual inspection of the detected spots. For tracking, the 'Autoregressive motion' algorithm was used with a maximum linking distance of 1.13 μm and a maximum gap size of 1 frame. Tracks with a number of spots <5 were not included. Using the track displacement length filter in Imaris, the number of tracks with track displacement length >1 μm were counted and divided by the total number of tracks to get the fractions plotted in *Figures 3 and 7*. Vectors generated in Imaris were used to manually determine the location and direction of tracks with track displacement length >1 μm. For the mean squared displacement (MSD) analysis, the xyzt coordinates were exported in 'csv' format for all tracks in a timelapse. The MSD analysis was done using the prewritten MATLAB class MSDanalyzer (*Tarantino et al., 2014*). MSDanalyzer calculates the mean squared displacement for each track, curve-fits the MSD vs time, and provides the value of the exponent α. The first 25% of the MSD vs time graph was used for curve-fitting. Tracks with the number of spots <10 were removed to ensure the accuracy of the MSD analysis. Fusion events between ribbons and precursors were scored manually in these timelapses.

## Statistics

All data shown are mean ± standard error of the mean (SEM) unless stated otherwise. All experiments were compiled from data acquired on at least two independent days from different clutches. All replicates were biological–distinct animals and cells. Wild-type animals were selected at random for drug treatments. Datasets were excluded if there was excessive x, y, or, z drift. In all datasets, dot plots represent the 'n.' N represents either the number of neuromasts, hair cells, synapses, or puncta as stated in the legends. For all zebrafish experiments, a minimum of three animals and six neuromasts were examined. Sample sizes were selected to avoid Type 2 errors. All statistical analyses were performed using Prism 10 software (GraphPad). A D'Agostino-Pearson normality test was used to test for normal distributions. To test for statistical significance between two samples, either unpaired t-tests (normally distributed data) or Wilcoxon or Mann-Whitney tests (non-normally distributed data) were used. For multiple comparisons, a one-way ANOVA was used. A p-value less than 0.05 was considered significant.

## Acknowledgements

We thank Drs Juan Angueyra and Katie Drerup for their comments on our manuscript. This work was supported by the National Institute on Deafness and Other Communication Disorders (NIDCD) Intramural Research Program Grant 1ZIADC000085-01 (KK) and Project B08 of the Collaborative Research

Center 889 '*Cellular Mechanisms of Sensory Processing*' of the German Research Foundation (MU, awarded to Christian Vogl).

## Additional information

### Funding

| Funder | Grant reference number | Author |
|---|---|---|
| National Institute on Deafness and Other Communication Disorders | 1ZIADC000085-01 | Saman Hussain Katherine Pinter Hiu-Tung Wong Katie S Kindt |
| Deutsche Forschungsgemeinschaft | Project B08 of the Collaborative Research Center 889 | Mara Uhl |

The funders had no role in study design, data collection and interpretation, or the decision to submit the work for publication.

### Author contributions

Saman Hussain, Conceptualization, Data curation, Software, Formal analysis, Investigation, Writing – original draft, Writing – review and editing; Katherine Pinter, Formal analysis, Investigation, Methodology, Writing – review and editing; Mara Uhl, Conceptualization, Formal analysis, Investigation; Hiu-Tung Wong, Conceptualization, Formal analysis, Investigation, Methodology; Katie S Kindt, Conceptualization, Resources, Data curation, Software, Formal analysis, Supervision, Funding acquisition, Validation, Investigation, Visualization, Methodology, Writing – original draft, Project administration, Writing – review and editing

### Author ORCIDs

Katie S Kindt ⬡ https://orcid.org/0000-0002-1065-8215

### Ethics

Zebrafish (Danio rerio) were bred and cared for at the National Institutes of Health (NIH). This research was approved by the NINDS/NIDCD/NCCIH animal care and use committee (ACUC) under animal study protocol #1362-13.

Reviewer #2 (Public review): https://doi.org/10.7554/eLife.98119.3.sa1
Reviewer #3 (Public review): https://doi.org/10.7554/eLife.98119.3.sa2
Author response https://doi.org/10.7554/eLife.98119.3.sa3

## Additional files

### Supplementary files

MDAR checklist

### Data availability

All data and code generated and used in this paper are available on Dryad: https://doi.org/10.5061/dryad.crjdfn3gg.

The following dataset was generated:

| Author(s) | Year | Dataset title | Dataset URL | Database and Identifier |
|---|---|---|---|---|
| Hussain S, Pinter K, Wong H, Uhl M, Kindt KS | 2025 | Microtubule networks in zebrafish hair cells facilitate presynapse transport and fusion during development | https://doi.org/10.5061/dryad.crjdfn3gg | Dryad Digital Repository, 10.5061/dryad.crjdfn3gg |

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
