## [Editor Report · eLife Assessment]

This **fundamental** study provides new insights into the maturation of ribbon synapses in zebrafish neuromast hair cells. Live-cell imaging and pharmacological and genetic manipulations together provide **compelling** evidence that the formation of this synaptic organelle is a dynamic process involving the fusion of presynaptic elements and microtubule transport, though the evidence that ribbon precursors move in a directed motion toward the active zone is less persuasive. These findings will be of interest to neuroscientists studying synapse formation and function and should inspire further research into the molecular basis for synaptic ribbon maturation.

---

## [Referee Report · Reviewer #2 (Public review)]

Summary:

In this manuscript, the authors set out to resolve a long-standing mystery in the field of sensory biology - how large, presynaptic bodies called "ribbon synapses" migrate to the basolateral end of hair cells. The ribbon synapse is found in sensory hair cells and photoreceptors, and is a critical structural feature of a readily releasable pool of glutamate that excites postsynaptic afferent neurons. For decades, we have known these structures exist, but the mechanisms that control how ribbon synapses coalesce at the bottom of hair cells is not well understood. The authors addressed this question by leveraging the highly-tractable zebrafish lateral line neuromast, which exhibits a small number of visible hair cells, easily observed in time-lapse imaging. The approach combined genetics, pharmacological manipulations, high-resolution imaging and careful quantifications. The manuscript commences with a developmental time course of ribbon synapse development, characterizing both immature and mature ribbon bodies (defined by position in the hair cell, apical vs. basal). Next, the authors show convincing (and frankly mesmerizing) imaging data of plus end-directed microtubule trafficking toward the basal end of the hair cells, and data highlighting the directed motion of ribbon bodies. The authors then use a series of pharmacological and genetic manipulations showing the role of microtubule stability and one particular kinesin (Kif1aa) in the transport and fusion of ribbon bodies, which is presumably all prerequisite for hair cell synaptic transmission. The data suggest that microtubules and their stability is necessary for normal numbers of mature ribbons, and that Kif1aa is likely required for fusion events associated with ribbon maturation. Overall, the data provide a new and interesting story on ribbon synapse dynamics.

Strengths:

(1) The manuscript offers comprehensive Introduction and Discussion sections that will inform generalists and specialists.

(2) The use of Airyscan imaging in living samples to view and measure microtubule and ribbon dynamics in vivo represents a strength. With the rigorous quantification and thoughtful analyses, the authors generate datasets often only gotten in cultured cells or more diminutive animal models (e.g., *C. elegans*).

(3) The number of biological replicates and the statistical analyses are strong. The combination of pharmacology and genetic manipulations also represents strong rigor.

(4) One of the most important strengths is that the manuscript and data spur on other questions - namely, do (or how do) ribbon bodies attach to Kinesin proteins? Also, and as noted in the Discussion, do hair cell activity and subsequent intracellular calcium rises facilitate ribbon transport/fusion.

---

## [Referee Report · Reviewer #3 (Public review)]

Summary:

The manuscript uses live imaging to study the role of microtubules in the movement of ribeye aggregates in neuromast hair cells in zebrafish. The main findings are that

(1) Ribeye aggregates, assumed to be ribbon precursors, move in a directed motion toward the active zone;

(2) Disruption of microtubules and kif1aa increases the number of ribeye aggregates and decreases the number of mature synapses.

The evidence for point 2 is compelling, while the evidence for point 1 is less convincing. In particular, the directed motion conclusion is dependent upon fitting of mean squared displacement that can be prone to error and variance to do stochasticity, which is not accounted for in the analysis. Only a small subset of the aggregates meet this criteria and one wonders whether the focus on this subset misses the bigger picture of what is happening with the majority of spots.

Strengths:

(1) The effects of Kif1aa removal and nocodozole on ribbon precursor number and size is convincing and novel.

(2) The live imaging of Ribeye aggregate dynamics provides interesting insight into ribbon formation. The movies showing fusion of ribeye spots are convincing and the demonstrated effects of nocodozole and kif1aa removal on the frequency of these events is novel.

(3) The effect of nocodozole and kif1aa removal on precursor fusion is novel and interesting.

(4) The quality of the data is extremely high and the results are interesting.

Weaknesses:

(1) To image ribeye aggregates, the investigators overexpressed Ribeye-a TAGRFP under control of a MyoVI promoter. While it is understandable why they chose to do the experiments this way, expression is not under the same transcriptional regulation as the native protein and some caution is warranted in drawing some conclusions. For example, the reduction in the number of puncta with maturity may partially reflect regulation of the MyoVI promoter with hair cell maturity. Similarly, it is unknown whether overexpression has the potential to saturate binding sites (for example to motors), which could influence mobility. In the revised manuscript, the authors provide evidence to suggest that overexpression is not at unreasonably high levels, which is reasonable. However, I think it remains important to think of these caveats while reading the paper--especially keeping in mind that expression timing is undoubtedly influenced by the transcriptional control of the exogenous promoter .

(2) The examples of punctae colocalizing with microtubules look clear (fig 1 F-G), but the presentation is anecdotal. It would be better and more informative, if quantified.

(3) It appears that any directed transport may be rare. Simply having an alpha >1 is not sufficient to declare movement to be directed (motor driven transport typically has an alpha approaching 2). Due to randomness of a random walk and errors in fits in imperfect data will yield some spread in movement driven by Brownian motion. Many of the tracks in figure 3H look as thought they might be reasonably fit by a straight line (i.e. alpha = 1).

(4) The "directed motion" shown here does not really resemble motor driven transport observed in other systems (axonal transport, for example) even in the subset that have been picked out as examples here. While the role for microtubules and kif1aa in synapse maturation is strong, it seems likely that this role may be something non-canonical (which would be interesting). In the revision, the authors do an excellent job of considering the issues brought up in point 3 and 4. While perhaps no longer a weakness, I am leaving the critiques here for context for the readers to consider. The added taxol results may not completely settle the issue, but are interesting and provide important information.

---

## [Author Response]

The following is the authors’ response to the original reviews

**Public Reviews:**

**Reviewer #1 (Public Review):**
Summary:The manuscript by Hussain and collaborators aims at deciphering the microtubule-dependent ribbon formation in zebrafish hair cells. By using confocal imaging, pharmacology tools, and zebrafish mutants, the group of Katie Kindt convincingly demonstrated that ribbon, the organelle that concentrates glutamate-filled vesicles at the hair cell synapse, originates from the fusion of precursors that move along the microtubule network. This study goes hand in hand with a complementary paper (Voorn et al.) showing similar results in mouse hair cells.Strengths:This study clearly tracked the dynamics of the microtubules, and those of the microtubule-associated ribbons and demonstrated fusion ribbon events. In addition, the authors have identified the critical role of kinesin Kif1aa in the fusion events. The results are compelling and the images and movies are magnificent.Weaknesses:The lack of functional data regarding the role of Kif1aa. Although it is difficult to probe and interpret the behavior of zebrafish after nocodazole treatment, I wonder whether deletion of kif1aa in hair cells may result in a functional deficit that could be easily tested in zebrafish?

We have examined functional deficits in kif1aa mutants in another paper that was recently accepted: David et al. 2024. https://pubmed.ncbi.nlm.nih.gov/39373584/

In David et al., we found that in addition to a subtle role in ribbon fusion during development, Kif1aa plays a major role in enriching glutamate-filled synaptic vesicles at the presynaptic active zone of mature hair cells. In kif1aa mutants, synaptic vesicles are no longer enriched at the hair cell base, and there is a reduction in the number of synaptic vesicles associated with presynaptic ribbons. Further, we demonstrated that kif1aa mutants also have functional defects including reductions in spontaneous vesicle release (from hair cells) and evoked postsynaptic calcium responses. Behaviorally, kif1aa mutants exhibit impaired rheotaxis, indicating defects in the lateral-line system and an inability to accurately detect water flow. Because our current paper focuses on microtubule-associated ribbon movement and dynamics early in hair-cell development, we have only discussed the effects of Kif1aa directly related to ribbon dynamics during this time window. In our revision, we have referenced this recent work. Currently it is challenging to disentangle how the subtle defects in ribbon formation in kif1aa mutants contribute to the defects we observe in ribbon-synapse function.

Added to results:

“Recent work in our lab using this mutant has shown that Kif1aa is responsible for enriching glutamate-filled vesicles at the base of hair cells. In addition this work demonstrated that loss of Kif1aa results in functional defects in mature hair cells including a reduction in evoked post-synaptic calcium responses (David et al., 2024). We hypothesized that Kif1aa may also be playing an earlier role in ribbon formation.”

Impact:The synaptogenesis in the auditory sensory cell remains still elusive. Here, this study indicates that the formation of the synaptic organelle is a dynamic process involving the fusion of presynaptic elements. This study will undoubtedly boost a new line of research aimed at identifying the specific molecular determinants that target ribbon precursors to the synapse and govern the fusion process.
**Reviewer #2 (Public Review):**
Summary:In this manuscript, the authors set out to resolve a long-standing mystery in the field of sensory biology - how large, presynaptic bodies called "ribbon synapses" migrate to the basolateral end of hair cells. The ribbon synapse is found in sensory hair cells and photoreceptors, and is a critical structural feature of a readily-releasable pool of glutamate that excites postsynaptic afferent neurons. For decades, we have known these structures exist, but the mechanisms that control how ribbon synapses coalesce at the bottom of hair cells are not well understood. The authors addressed this question by leveraging the highly-tractable zebrafish lateral line neuromast, which exhibits a small number of visible hair cells, easily observed in time-lapse imaging. The approach combined genetics, pharmacological manipulations, high-resolution imaging, and careful quantifications. The manuscript commences with a developmental time course of ribbon synapse development, characterizing both immature and mature ribbon bodies (defined by position in the hair cell, apical vs. basal). Next, the authors show convincing (and frankly mesmerizing) imaging data of plus end-directed microtubule trafficking toward the basal end of the hair cells, and data highlighting the directed motion of ribbon bodies. The authors then use a series of pharmacological and genetic manipulations showing the role of microtubule stability and one particular kinesin (Kif1aa) in the transport and fusion of ribbon bodies, which is presumably a prerequisite for hair cell synaptic transmission. The data suggest that microtubules and their stability are necessary for normal numbers of mature ribbons and that Kif1aa is likely required for fusion events associated with ribbon maturation. Overall, the data provide a new and interesting story on ribbon synapse dynamics.Strengths:(1) The manuscript offers a comprehensive Introduction and Discussion sections that will inform generalists and specialists.(2) The use of Airyscan imaging in living samples to view and measure microtubule and ribbon dynamics in vivo represents a strength. With rigorous quantification and thoughtful analyses, the authors generate datasets often only obtained in cultured cells or more diminutive animal models (e.g., *C. elegans*).(3) The number of biological replicates and the statistical analyses are strong. The combination of pharmacology and genetic manipulations also represents strong rigor.(4) One of the most important strengths is that the manuscript and data spur on other questions - namely, do (or how do) ribbon bodies attach to Kinesin proteins? Also, and as noted in the Discussion, do hair cell activity and subsequent intracellular calcium rises facilitate ribbon transport/fusion?

These are important strengths and as stated we are currently investigating what other kinesins and adaptors and adaptor’s transport ribbons. We have ongoing work examining how hair-cell activity impacts ribbon fusion and transport!

Weaknesses:(1) Neither the data or the Discussion address a direct or indirect link between Kinesins and ribbon bodies. Showing Kif1aa protein in proximity to the ribbon bodies would add strength.

This is a great point. Previous immunohistochemistry work in mice demonstrated that ribbons and Kif1a colocalize in mouse hair cells (Michanski et al, 2019). Unfortunately, the antibody used in study work did not work in zebrafish. To further investigate this interaction, we also attempted to create a transgenic line expressing a fluorescently tagged Kif1aa to directly visualize its association with ribbons in vivo. At present, we were unable to detect transient expression of Kif1aa-GFP or establish a transgenic line using this approach. While we will continue to work towards understanding whether Kif1aa and ribbons colocalize in live hair cells, currently this goal is beyond the scope of this paper. In our revision we discuss this caveat.

Added to discussion:

“In addition, it will be useful to visualize these kinesins by fluorescently tagging them in live hair cells to observe whether they associate with ribbons.”

(2) Neither the data or Discussion address the functional consequences of loss of Kif1aa or ribbon transport. Presumably, both manipulations would reduce afferent excitation.

Excellent point. Please see the response above to Reviewer #1 public response weaknesses.

(3) It is unknown whether the drug treatments or genetic manipulations are specific to hair cells, so we can't know for certain whether any phenotypic defects are secondary.

This is correct and a caveat of our Kif1aa and drug experiments. In our recently published work, we confirmed that Kif1aa is expressed in hair cells and neurons, while kif1ab is present just is neurons. Therefore, it is likely that the ribbon formation defects in kif1aa mutants are restricted to hair cells. We added this expression information to our results:

“ScRNA-seq in zebrafish has demonstrated widespread co-expression of kif1ab and kif1aa mRNA in the nervous system. Additionally, both scRNA-seq and fluorescent in situ hybridization have revealed that pLL hair cells exclusively express kif1aa mRNA (David et al., 2024; Lush et al., 2019; Sur et al., 2023).”

Non-hair cell effects are a real concern in our pharmacology experiments. To mitigate this in our pharmacological experiments, we have performed drug treatments at 3 different timescales: long-term (overnight), short-term (4 hr) and fast (30 min) treatments. The fast experiments were done after 30 min nocodazole drug treatment, and after this treatment we observed reduced directional motion and fusions. This fast drug treatment should not incur any long-term changes or developmental defects as hair-cell development occurs over 12-16 hrs. However, we acknowledge that drug treatments could have secondary phenotypic effects or effects that are not hair-cell specific. In our revision, we discuss these issues.

Added to discussion:

“Another important consideration is the potential off-target effects of nocodazole. Even at non-cytotoxic doses, nocodazole toxicity may impact ribbons and synapses independently of its effects on microtubules. While this is less of a concern in the short- and medium-term experiments (30-70 min and 4 hr), long-term treatments (16 hrs) could introduce confounding effects. Additionally, nocodazole treatment is not hair cell-specific and could disrupt microtubule organization within afferent terminals as well. Thus, the reduction in ribbon-synapse formation following prolonged nocodazole treatment may result from microtubule disruption in hair cells, afferent terminals, or a combination of the two.”

**Reviewer #3 (Public Review):**
Summary:The manuscript uses live imaging to study the role of microtubules in the movement of ribeye aggregates in neuromast hair cells in zebrafish. The main findings are that(1) Ribeye aggregates, assumed to be ribbon precursors, move in a directed motion toward the active zone;(2) Disruption of microtubules and kif1aa increases the number of ribeye aggregates and decreases the number of mature synapses.The evidence for point 2 is compelling, while the evidence for point 1 is less convincing. In particular, the directed motion conclusion is dependent upon fitting of mean squared displacement that can be prone to error and variance to do stochasticity, which is not accounted for in the analysis. Only a small subset of the aggregates meet this criteria and one wonders whether the focus on this subset misses the bigger picture of what is happening with the majority of spots.Strengths:(1) The effects of Kif1aa removal and nocodozole on ribbon precursor number and size are convincing and novel.(2) The live imaging of Ribeye aggregate dynamics provides interesting insight into ribbon formation. The movies showing the fusion of ribeye spots are convincing and the demonstrated effects of nocodozole and kif1aa removal on the frequency of these events is novel.(3) The effect of nocodozole and kif1aa removal on precursor fusion is novel and interesting.(4) The quality of the data is extremely high and the results are interesting.Weaknesses:(1) To image ribeye aggregates, the investigators overexpressed Ribeye-a TAGRFP under the control of a MyoVI promoter. While it is understandable why they chose to do the experiments this way, expression is not under the same transcriptional regulation as the native protein, and some caution is warranted in drawing some conclusions. For example, the reduction in the number of puncta with maturity may partially reflect the regulation of the MyoVI promoter with hair cell maturity. Similarly, it is unknown whether overexpression has the potential to saturate binding sites (for example motors), which could influence mobility.

We agree that overexpression of transgenes under using a non-endogenous promoter in transgenic lines is an important consideration. Ideally, we would do these experiments with endogenously expressed fluorescent proteins under a native promoter. However, this was not technically possible for us. The decrease in precursors is likely not due to regulation by the myo6a promoter. Although the myo6a promoter comes on early in hair cell development, the promoter only gets stronger as the hair cells mature. This would lead to a continued increase rather than a decrease in puncta numbers with development.

Protein tags such as tagRFP always have the caveat of impacting protein function. This is in partly why we complemented our live imaging with analyses in fixed tissue without transgenes (kif1aa mutants and nocodazole/taxol treatments).

In our revision, we did perform an immunolabel on *myo6b:riba-tagRFP* transgenic fish and found that Riba-tagRFP expression did not impact ribbon synapse numbers or ribbon size. This analysis argues that the transgene is expressed at a level that does not impact ribbon synapses. This data is summarized in Figure 1-S1.

Added to the results:

“Although this latter transgene expresses Riba-TagRFP under a non-endogenous promoter, neither the tag nor the promoter ultimately impacts cell numbers, synapse counts, or ribbon size (Figure 1-S1A-E).”

Added to methods:

“*Tg(myo6b:ctbp2a-TagRFP)idc11Tg* reliably labels mature ribbons, similar to a pan-CTBP immunolabel at 5 dpf (Figure 1-S1B). This transgenic line does not alter the number of hair cells or complete synapses per hair cell (Figure 1-S1A-D). In addition, *myo6b:ctbp2a-TagRFP* does not alter the size of ribbons (Figure 1-S1E).”

(2) The examples of punctae colocalizing with microtubules look clear (Figures 1 F-G), but the presentation is anecdotal. It would be better and more informative, if quantified.

We did attempt a co-localization analysis between microtubules and ribbons but did not move forward with it due to several issues:

(1) Hair cells have an extremely crowded environment, especially since the nucleus occupies the majority of the cell. All proteins are pushed together in the small space surrounding the nucleus and ultimately, we found that co-localization analyses were not meaningful because the distances were too small.

(2) We also attempted to segment microtubules in these images and quantify how many ribbons were associated with microtubules, but 3D microtubule segmentation was not accurate in hair cells due to highly varying filament intensities, filament dynamics and the presence of diffuse cytoplasmic tubulin signal.

Because of these challenges we concluded the best evidence of ribbon-microtubule association is through visualization of ribbons and their association with microtubules over time (in our timelapses). We see that ribbons localize to microtubules in all our timelapses, including the examples shown (Movies S2-S10). The only instance of ribbon dissociation it when ribbons switch from one filament to another. We did not observe free-floating ribbons in our study.

(3) It appears that any directed transport may be rare. Simply having an alpha >1 is not sufficient to declare movement to be directed (motor-driven transport typically has an alpha approaching 2). Due to the randomness of a random walk and errors in fits in imperfect data will yield some spread in movement driven by Brownian motion. Many of the tracks in Figure 3H look as though they might be reasonably fit by a straight line (i.e. alpha = 1).(4) The "directed motion" shown here does not really resemble motor-driven transport observed in other systems (axonal transport, for example) even in the subset that has been picked out as examples here. While the role of microtubules and kif1aa in synapse maturation is strong, it seems likely that this role may be something non-canonical (which would be interesting).

Yes, it is true, that directed transport of ribbon precursors is relatively rare. Only a small subset of the ribbon precursors moves directionally (α > 1, 20 %) or have a displacement distance > 1 µm (36 %) during the time windows we are imaging. The majority of the ribbons are stationary. To emphasize this result we have added bar graphs to Figure 3I,K to illustrate this result and state the numbers behind this result more clearly.

“Upon quantification, 20.2 % of ribbon tracks show α > 1, indicative of directional motion, but the majority of ribbon tracks (79.8 %) show α < 1, indicating confinement on microtubules (Figure 3I, n = 10 neuromasts, 40 hair cells, and 203 tracks).

To provide a more comprehensive analysis of precursor movement, we also examined displacement distance (Figure 3J). Here, as an additional measure of directed motion, we calculated the percent of tracks with a cumulative displacement > 1 µm. We found 35.6 % of tracks had a displacement > 1 µm (Figure 3K; n = 10 neuromasts, 40 hair cells, and 203 tracks).”

We cannot say for certain what is happening with the stationary ribbons, but our hypothesis is that these ribbons eventually exhibit directed motion sufficient to reach the active zone. This idea is supported by the fact that we see ribbons that are stationary begin movement, and ribbons that are moving come to a stop during the acquisition of our timelapses (Movies S4 and S5). It is possible that ribbons that are stationary may not have enough motors attached, or there may be a ‘seeding’ phase where Ribeye aggregates are condensing on the ribbon.

We also reexamined our MSD a values as the a values we observed in hair cells were lower than those seen canonical motor-driven transport (where a approaches 2). One reason for this difference may arise from the dynamic microtubule network in developing hair cells, which could affect directional ribbon movement. In our revision we plotted the distribution of a values which confirmed that in control hair cells, the majority of the a values we see are typically less than 2 (Figure 7-S1A). Interestingly we also compared the distribution a values between control and taxol-treated hair cells, where the microtubule network is more stable, and found that the distribution shifted towards higher a values (Figure 7-S1A). We also plotted only ‘directional’ tracks (with a > 1) and observed significantly higher a values in taxol-treated hair cells (Figure 7-S1B). This is an interesting result which indicates that although the proportion of directional tracks (with a > 1) is not significantly different between control and taxol-treated hair cells (which could be limited by the number of motor/adapter proteins), the ribbons that move directionally do so with greater velocities when the microtubules are more stable. This supports our idea that the stability of the microtubule network could be why ribbon movement does not resemble canonical motor transport. This analysis is presented as a new figure (Figure 7-S1A-B) and is referred to in the text in the results and the discussion.

Results:

“Interestingly, when we examined the distribution of α values, we observed that taxol treatment shifted the overall distribution towards higher α a values (Figure 7-S1A). In addition, when we plotted only tracks with directional motion (α > 1), we found significantly higher α values in hair cells treated with taxol compared to controls (Figure 7-S1B). This indicates that in taxol-treated hair cells, where the microtubule network is stabilized, ribbons with directional motion have higher velocities.”

Discussion:

“Our findings indicate that ribbons and precursors show directed motion indicative of motor-mediated transport (Figure 3 and 7). While a subset of ribbons moves directionally with α values > 1, canonical motor-driven transport in other systems, such as axonal transport, can achieve even higher α values approaching 2 (Bellotti et al., 2021; Corradi et al., 2020). We suggest that relatively lower α values arise from the highly dynamic nature of microtubules in hair cells. In axons, microtubules form stable, linear tracks that allow kinesins to transport cargo with high velocity. In contrast, the microtubule network in hair cells is highly dynamic, particularly near the cell base. Within a single time frame (50-100 s), we observe continuous movement and branching of these networks. This dynamic behavior adds complexity to ribbon motion, leading to frequent stalling, filament switching, and reversals in direction. As a result, ribbon transport appears less directional than the movement of traditional motor cargoes along stable axonal filaments, resulting in lower α values compared to canonical motor-mediated transport. Notably, treatment with taxol, which stabilizes microtubules, increased α values to levels closer to those observed in canonical motor-driven transport (Figure 7-S1). This finding supports the idea that the relatively lower α values in hair cells are a consequence of a more dynamic microtubule network. Overall, this dynamic network gives rise to a slower, non-canonical mode of transport.”

(5) The effect of acute treatment with nocodozole on microtubules in movie 7 and Figure 6 is not obvious to me and it is clear that whatever effect it has on microtubules is incomplete.

When using nocodazole, we worked to optimize the concentration of the drug to minimize cytotoxicity, while still being effective. While the more stable filaments at the cell apex remain largely intact after nocodazole treatment, there are almost no filaments at the hair cell base, which is different from the wild-type hair cells. In addition, nocodazole-treated hair cells have more cytoplasmic YFP-tubulin signal compared to wild type. We have clarified this in our results. To better illustrate the effect of nocodazole and taxol we have also added additional side-view images of hair cells expressing YFP-tubulin (Figure 4-S1F-G), that highlight cytoplasmic YFP-tubulin and long, stabilized microtubules after 3-4 hr treatment with nocodazole and taxol respectively. In these images we also point out microtubules at the apical region of hair cells that are very stable and do not completely destabilize with nocodazole treatment at concentrations that are tolerable to hair cells.

“We verified the effectiveness of our in vivo pharmacological treatments using either 500 nM nocodazole or 25 µM taxol by imaging microtubule dynamics in pLL hair cells (myo6b:YFP-tubulin). After a 30-min pharmacological treatment, we used Airyscan confocal microscopy to acquire timelapses of YFP-tubulin (3 µm z-stacks, every 50-100 s for 30-70 min, Movie S8). Compared to controls, 500 nM nocodazole destabilized microtubules (presence of depolymerized YFP-tubulin in the cytosol, see arrows in Figure 4-S1F-G) and 25 µM taxol dramatically stabilized microtubules (indicated by long, rigid microtubules, see arrowheads in Figure 4-S1F,H) in pLL hair cells. We did still observe a subset of apical microtubules after nocodazole treatment, indicating that this population is particularly stable (see asterisks in Figure 4-S1F-H).”

To further address concerns about verifying the efficacy of nocodazole and taxol treatment on microtubules, we added a quantification of our immunostaining data comparing the mean acetylated-a-tubulin intensities between control, nocodazole and taxol-treated hair cells. Our results show that nocodazole treatment reduces the mean acetylated-a-tubulin intensity in hair cells. This is included as a new figure (Figure 4-S1D-E) and this result is referred to in the text. To better illustrate the effect of nocodazole and taxol we have also added additional side-view images of hair cells after overnight treatment with nocodazole and taxol (Figure 4-S1A-C).

“After a 16-hr treatment with 250 nM nocodazole we observed a decrease in acetylated-a-tubulin label (qualitative examples: Figure 4A,C, Figure 4-S1A-B). Quantification revealed significantly less mean acetylated-a-tubulin label in hair cells after nocodazole treatment (Figure 4-S1D). Less acetylated-a-tubulin label indicates that our nocodazole treatment successfully destabilized microtubules.”

“Qualitatively more acetylated-a-tubulin label was observed after treatment, indicating that our taxol treatment successfully stabilized microtubules (qualitative examples: Figure 4-S1A,C). Quantification revealed an overall increase in mean acetylated-a-tubulin label in hair cells after taxol treatment, but this increase did not reach significance (Figure 4-S1E).”

**Recommendations for the authors:**

**Reviewer #1 (Recommendations For The Authors):**
(1) The manuscript is fairly dense. For instance, some information is repeated (page 3 ribbon synapses form along a condensed timeline in zebrafish hair cells: 12-18 hrs, and on .page 5. These hair cells form 3-4 ribbon synapses in just 12-18 hrs). Perhaps, the authors could condense some of the ideas? The introduction could be shortened.

We have eliminated this repeated text in our revision. We have shortened the introduction 1275 to 1038 words (with references)

(2) The mechanosensory structure on page 5 is not defined for readers outside the field.

Great point, we have added addition information to define this structure in the results:

“We staged hair cells based on the development of the apical, mechanosensory hair bundle. The hair bundle is composed of actin-based stereocilia and a tubulin-based kinocilium. We used the height of the kinocilium (see schematic in Figure 1B), the tallest part of the hair bundle, to estimate the developmental stage of hair cells as described previously…”

(3) Figure 1E is quite interesting but I'd rather show Figure S1 B/C as they provide statistics. In addition, the authors define 4 stages : early, intermediate, late, and mature for counting but provide only 3 panels for representative examples by mixing late/mature.

We were torn about which ribbon quantification graph to show. Ultimately, we decided to keep the summary data in Figure 1E. This is primarily because the supplementary Figure will be adjacent to the main Figure in the Elife format, and the statistics will be easy to find and view.

Figure 1 now provides a representative image for both late and mature hair cells.

(4.) The ribbon that jumps from one microtubule to another one is eye-catching. Can the authors provide any statistics on this (e.g. percentage)?

Good point. In our revision, we have added quantification for these events. We observe 2.8 switching events per neuromast during our fast timelapses. This information is now in the text and is also shown in a graph in Figure 3-S1D.

“Third, we often observed that precursors switched association between neighboring microtubules (2.8 switching events per neuromast, n = 10 neuromasts; Figure 3-S1C-D, Movie S7).”

(5) With regard to acetyl-a-tub immunocytochemistry, I would suggest obtaining a profile of the fluorescence intensity on a horizontal plane (at the apical part and at the base).(6) Same issue with microtubule destruction by nocodazole. Can the authors provide fluorescence intensity measurements to convince readers of microtubule disruption for long and short-term application.

Regarding quantification of microtubule disruption using nocodazole and taxol. We did attempt to create profiles of the acetylated tubulin or YFP-tubulin label along horizontal planes at the apex and base, but the amount variability among cells and the angle of the cell in the images made this type of display and quantification challenging. In our revision we as stated above in our response to Reviewer #1’s public comment, we have added representative side-view images to show the disruptions to microtubules more clearly after short and long-term drug experiments (Figure 4-S1A-C, F-H). In addition, we quantified the reduction in acetylated tubulin label after overnight treatment with nocodazole and found the signal was significantly reduced (Figure 3-S1D-E). Unfortunately, we were unable to do a similar quantification due to the variability in YFP-tubulin intensity due to variations in mounting. The following text has been added to the results:

“Quantification revealed significantly less mean acetylated-a-tubulin label in hair cells after nocodazole treatment (Figure 4-S1D).”

“Quantification revealed an overall increase in mean acetylated-a-tubulin label in hair cells after taxol treatment, but this increase did not reach significance (Figure 4-S1A,C,E).”

(7) It is a bit difficult to understand that the long-term (overnight) microtubule destabilization leads to a reduction in the number of synapses (Figure 4F) whereas short-term (30 min) microtubule destabilization leads to the opposite phenotype with an increased number of ribbons (Figure 6G). Are these ribbons still synaptic in short-term experiments? What is the size of the ribbons in the short-term experiments? Alternatively, could the reduction in synapse number upon long-term application of nocodazole be a side-effect of the toxicity within the hair cell?

Agreed-this is a bit confusing. In our revision, we have changed our analyses, so the comparisons are more similar between the short- and long-term experiments–we examined the number of ribbons and precursor per cells (apical and basal) in both experiments (Changed the panel in Figure 4G, Figure 4-S2G and Figure 5G). In our live experiments we cannot be sure that ribbons are synaptic as we do not have a postsynaptic co-label. Also, we are unable to reliably quantify ribbon and precursor size in our live images due to variability in mounting. We have changed the text to clarify as follows:

Results:

“In each developing cell, we quantified the total number of Riba-TagRFP puncta (apical and basal) before and after each treatment. In our control samples we observed on average no change in the number of Riba-TagRFP puncta per cell (Figure 6G). Interestingly, we observed that nocodazole treatment led to a significant increase in the total number of Riba-TagRFP puncta after 3-4 hrs (Figure 6G). This result is similar to our overnight nocodazole experiments in fixed samples, where we also observed an increase in the number of ribbons and precursors per hair cell. In contrast to our 3-4 hr nocodazole treatment, similar to controls, taxol treatment did not alter the total number of Riba-TagRFP puncta over 3-4 hrs (Figure 6G). Overall, our overnight and 3-4 hr pharmacology experiments demonstrate that microtubule destabilization has a more significant impact on ribbon numbers compared to microtubule stabilization.”

Discussion:

“Ribbons and microtubules may interact during development to promote fusion, to form larger ribbons. Disrupting microtubules could interfere with this process, preventing ribbon maturation. Consistent with this, short-term (3-4 hr) and long-term (overnight) nocodazole increased ribbon and precursor numbers (Figure 6AG; Figure 4G), suggesting reduced fusion. Long-term treatment (overnight) resulted in a shift toward smaller ribbons (Figure 4H-I), and ultimately fewer complete synapses (Figure 4F).”

Nocodazole toxicity: in response to Reviewer # 2’s public comment we have added the following text in our discussion:

Discussion:

“Another important consideration is the potential off-target effects of nocodazole. Even at non-cytotoxic doses, nocodazole toxicity may impact ribbons and synapses independently of its effects on microtubules. While this is less of a concern in the short- and medium-term experiments (30 min to 4 hr), long-term treatments (16 hrs) could introduce confounding effects. Additionally, nocodazole treatment is not hair cell-specific and could disrupt microtubule organization within afferent terminals as well. Thus, the reduction in ribbon-synapse formation following prolonged nocodazole treatment may result from microtubule disruption in hair cells, afferent terminals, or a combination of the two.”

(8) Does ribbon motion depend on size or location?

It is challenging to reliability quantify the actual area of precursors in our live samples, as there is variability in mounting and precursors are quite small. But we did examine the location of ribbon precursors (using tracks > 1 µm as these tracks can easily be linked to cell location in Imaris) with motion in the cell. We found evidence of ribbons with tracks > 1 µm throughout the cell, both above and below the nucleus. This is now plotted in Figure 3M. We have also added the following test to the results:

“In addition, we examined the location of precursors within the cell that exhibited displacements > 1 µm. We found that 38.9 % of these tracks were located above the nucleus, while 61.1 % were located below the nucleus (Figure 3M).”

Although this is not an area or size measurement, this result suggests that both smaller precursors that are more apical, and larger precursors/ribbons that are more basal all show motion.

(9) The fusion event needs to be analyzed in further detail: when one ribbon precursor fuses with another one, is there an increase in size or intensity (this should follow the law of mass conservation)? This is important to support the abstract sentence "ribbon precursors can fuse together on microtubules to form larger ribbons".

As mentioned above it is challenging accurately estimate the absolute size or intensity of ribbon precursors in our live preparation. But we did examine whether there is a relative increase in area after ribbon fuse. We have plotted the change in area (within the same samples) for the two fusion events in shown in Figure 8-S1A-B. In these examples, the area of the puncta after fusion is larger than either of the two precursors that fuse. Although the areas are not additive, these plots do provide some evidence that fusion does act to form larger ribbons. To accompany these plots, we have added the following text to the results:

“Although we could not accurately measure the areas of precursors before and after fusion, we observed that the relative area resulting from the fusion of two smaller precursors was greater than that of either precursor alone. This increase in area suggests that precursor fusion may serve as a mechanism for generating larger ribbons (see examples: Figure 8-S1A-B).”

Because we were unable to provide more accurate evidence of precursor fusion resulting in larger ribbons, we have removed this statement from our abstract and lessened our claims elsewhere in the manuscript.

(10) The title in Figure 8 is a bit confusing. If fusion events reflect ribbon precursors fusion, it is obvious it depends on ribbon precursors. I'd like to replace this title with something like "microtubules and kif1aa are required for fusion events"

We have changed the figure title as suggested, good idea.

**Reviewer #2 (Recommendations For The Authors):**
(1) Figure 1C. The purple/magenta colors are hard to distinguish.

We have made the magenta color much lighter in the Figure 1C to make it easier to distinguish purple and magenta.

(2) There are places where some words are unnecessarily hyphenated. Examples: live-imaging and hair-cell in the abstract, time-course in the results.

In our revision, we have done our best to remove unnecessary hyphens, including the ones pointed out here.

(3) Figure 4H and elsewhere - what is "area of Ribeye puncta?" Related, I think, in the Discussion the authors refer to "ribbon volume" on line 484. But they never measured ribbon volume so this needs to be clarified.

We have done best to clarify what is meant by area of Ribeye puncta in the results and the methods:

Results:

“We also observed that the average of individual Ribeyeb puncta (from 2D max-projected images) was significantly reduced compared to controls (Figure 4H). Further, the relative frequency of individual Ribeyeb puncta with smaller areas was higher in nocodazole treated hair cells compared to controls (Figure 4I).”

Methods:

“To quantify the area of each ribbon and precursor, images were processed in a FIJI ‘IJMacro_AIRYSCAN_simple3dSeg_ribbons only.ijm’ as previously described (Wong et al., 2019). Here each Airyscan z-stack was max-projected. A threshold was applied to each image, followed by segmentation to delineate individual Ribeyeb/CTBP puncta. The watershed function was used to separate adjacent puncta. A list of 2D objects of individual ROIs (minimum size filter of 0.002 μm2) was created to measure the 2D areas of each Ribeyeb/CTBP puncta.”

We did refer to ribbon volume once in the discussion, but volume is not reflected in our analyses, so we have removed this mention of volume.

(4) More validation data showing gene/protein removal for the crispants would be helpful.

Great suggestion. As this is a relatively new method, we have created a figure that outlines how we genotype each individual crispant animal analyzed in our study Figure 6-S1. In the methods we have also added the following information:

“fPCR fragments were run on a genetic analyzer (Applied Biosystems, 3500XL) using LIZ500 (Applied Biosystems, 4322682) as a dye standard. Analysis of this fPCR revealed an average peak height of 4740 a.u. in wild type, and an average peak height of 126 a.u. in kif1aa F0 crispants (Figure 6-S1). Any kif1aa F0 crispant without robust genomic cutting or a peak height > 500 a.u. was not included in our analyses.”

**Reviewer #3 (Recommendations For The Authors):**
Lines 208-209--should refer to the movie in the text.

Movie S1 is now referenced here.

It would be helpful if the authors could analyze and quantify the effect of nocodozole and taxol on microtubules (movie 7).

See responses above to Reviewer #1’s similar request.

Figure 7 caption says "500 mM" nocodozole.

Thank you, we have changed the caption to 500 nM.

One problem with the MSD analysis is that it is dependent upon fits of individual tracks that lead to inaccuracies in assigning diffusive, restricted, and directed motion. The authors might be able to get around these problems by looking at the ensemble averages of all the tracks and seeing how they change with the various treatments. Even if the effect is on a subset of ribeye spots, it would be reassuring to see significant effects that did not rely upon fitting.

We are hesitant to average the MSD tracks as not all tracks have the same number of time steps (ribbon moving in and out of the z-stack during the timelapse). This makes it challenging for us to look at the ensembles of all averages accurately, especially for the duration of the timelapse. This is the main reason why added another analysis, displacements > 1µm as another readout of directional motion, a measure that does not rely upon fitting.

The abstract states that directed movement is toward the synapse. The only real evidence for this is a statement in the results: "Of the tracks that showed directional motion, while the majority move to the cell base, we found that 21.2 % of ribbon tracks moved apically." A clearer demonstration of this would be to do the analysis of Figure 2G for the ribeye aggregates.

If was not possible to do the same analysis to ribbon tracks that we did for the EB3-GFP analysis in Figure 2. In Figure 2 we did a 2D tracking analysis and measured the relative angles in 2D. In contrast, the ribbon tracking was done in 3D in Imaris not possible to get angles in the same way. Further the MSD analysis was outside of Imaris, making it extremely difficult to link ribbon trajectories to the 3D cellular landscape in Imaris. Instead, we examined the direction of the 3D vectors in Imaris with tracks > 1µm and determined the direction of the motion (apical, basal or undetermined). For clarity, this data is now included as a bar graph in Figure 3L. In our results, we have clarified the results of this analysis:

“To provide a more comprehensive analysis of precursor movement, we also examined displacement distance (Figure 3J). Here, as an additional measure of directed motion, we calculated the percent of tracks with a cumulative displacement > 1 µm. We found 35.6 % of tracks had a displacement > 1 µm (Figure 3K; n = 10 neuromasts, 40 hair cells and 203 tracks). Of the tracks with displacement > 1 µm, the majority of ribbon tracks (45.8 %) moved to the cell base, but we also found a subset of ribbon tracks (20.8 %) that moved apically (33.4 % moved in an undetermined direction) (Figure 3L).”

Some more detail about the F0 crispants should be provided. In particular, what degree of cutting was observed and what was the criteria for robust cutting?

See our response to Reviewer 2 and the newly created Figure 6-S1.